# Remote Sensing Estimates of Time-Resolved HONO and NO$_2$ Emission Rates and Lifetimes in Wildfires

Carley D. Fredrickson[1†], Scott J. Janz[2], Lok N. Lamsal[2,3], Ursula A. Jongebloed[1], Joshua L. Laughner[4], Joel A. Thornton[1]

[1]Department of Atmospheric and Climate Science, University of Washington, Seattle, Washington 98195, United States
[2]NASA Goddard Space Flight Center, Greenbelt, Maryland 20771, United States
[3]Goddard Earth Sciences Technology and Research, University of Maryland, Baltimore County, Baltimore, Maryland, 21228, United States
[4]Jet Propulsion Laboratory, California Institute of Technology, Pasadena, California, 91109, United States

[†]Now at Ramboll Americas Engineering Solutions, Inc., Seattle, Washington 98164, United States

*Correspondence to*: Joel A. Thornton (joelt@uw.edu)

**Abstract.** Quantification of wildfire emissions is essential for comprehending and simulating the effects of wildfires on atmospheric chemical composition. Sub-orbital measurements of vertical column nitrous acid (HONO) and nitrogen dioxide (NO$_2$) were made during the Fire Influence on Regional to Global Environments and Air Quality (FIREX-AQ) field campaign using the GeoCAPE Airborne Simulator (GCAS) instrument aboard the NASA ER-2 aircraft. Emission rates and lifetimes of HONO and NO$_2$ from the Sheridan Fire were estimated by fitting exponentially modified Gaussians (EMGs) to line densities, a technique previously used to estimate urban and point source NO$_2$ emissions. As the EMG approach does not capture temporal changes in emissions and lifetimes due to time-varying fire behavior, we developed a Monte Carlo implementation of the Python Editable Chemical Atmospheric Numeric Solver (PECANS) model that includes diurnal fire radiative power (FRP) behavior. We assess the validity of a range of emission rate and lifetime combinations for both HONO and NO$_2$ as the fire evolves by comparing the resulting line density predictions to the line density observations. We find that our method results in emissions that are lower than top-down biomass burning emissions inventories and higher than bottom-up inventories. Our approach is applicable to interpreting time-resolved remotely sensed measurements of atmospheric trace gases such as those now becoming available with instruments aboard geo-stationary satellites such as the Tropospheric Emissions: Monitoring of Pollution (TEMPO) and the Geostationary Environment Monitoring Spectrometer (GEMS) instruments.

## 1 Introduction

Wildfires pose several risks to infrastructure, air quality, and climate. Wildfires emit particulate matter, reduced and oxidized nitrogen, carbon monoxide, and volatile organic compounds (VOCs) that degrade local and regional air quality and impact human health (Akagi et al., 2011; Andreae, 2019). The reactions of the compounds found in wildfire smoke additionally lead

to the creation of ozone and other hazardous air pollutants (Buysse et al., 2019; Jaffe and Wigder, 2012). The impact of wildfires can even extend past the troposphere, where pyrocumulonimbus clouds formed from wildfires can inject smoke particles into the stratosphere, enhancing chlorine activation and stratospheric ozone depletion, as well as surface cooling

and stratospheric heating (Bernath et al., 2022; Solomon et al., 2022, 2023; Ye et al., 2021). As the intensity of fires and burned area from fires in the United States and globally have had increasing trends over the past few decades and are predicted to increase in the future (Barbero et al., 2015; Burton et al., 2024; Cunningham et al., 2024; Dennison et al., 2014), it is important to understand how wildfire emissions change with fire properties and how these emissions impact local, regional, and global atmospheric composition.


There are two distinct approaches for estimating global fire emissions within biomass burning emission inventories: (1) a bottom-up approach that uses burned area and (2) a top-down approach that uses fire radiative power (FRP) as proxies for amount of material burned. Some examples of burned area inventories are the Global Fire Emissions Database (GFED; Giglio et al., 2013) and the Fire INventory from NCAR (FINN) (Wiedinmyer et al., 2011). Some examples of FRP-derived

inventories are the Quick Fire Emissions Dataset (QFED; Darmenov and da Silva, 2015) and the Global Fire Assimilation System (GFAS) (Kaiser et al., 2012). Both approaches ultimately rely on emission factors, which translate the biomass burned to the emitted mass of species constrained by *in situ* or remotely sensed observations. The amount of biomass burned is tied to satellite-measured surface properties, but fire emissions can also be estimated by measuring a fire's smoke plume within the atmosphere.


While historically used to measure the emissions and lifetimes of urban pollution (Goldberg et al., 2019; Laughner and Cohen, 2019), recently the exponentially modified Gaussian (EMG) approach has also been applied to wildfire smoke plumes detected by polar-orbiting satellites which make observations typically once a day (Griffin et al., 2021; Jin et al., 2021). The EMG is fit to a plume and the enhancement factor from the fitting procedure is directly linked to the emissions.

Other methods are also used to estimate emissions from satellites, such as the integrated mass enhancement (IME) method, which multiplies the wind speed by the integrated vertical column densities (VCDs), the cross-sectional flux method (CFM), which estimates emissions by averaging the flux through multiple cross sections perpendicular to the plume direction, and Chemical Transport Models (CTMs), which can predict emissions more accurately but are computationally expensive. While these other techniques are proven to be useful, this paper focuses on dissecting the EMG approach. Forming conclusions

using daily observations of fires, while a good starting point, does not capture the diurnal variability that fires exhibit (Wiggins et al., 2020). With the emergence of geostationary satellites reporting atmospheric composition, our understanding of the diurnal emissions of wildfires and the extent of their variability will be vastly improved.

Recent advances in remote sensing and *in situ* observations have enabled improved assessments of wildfire emissions of

reactive nitrogen (Gkatzelis et al., 2024; Lindaas et al., 2021; Theys et al., 2020). Nitrogen dioxide ($NO_2$) is routinely

measured by satellites and by air quality monitors and these observations have been used extensively to estimate nitrogen oxides ($NO_x$; $NO_x = NO + NO_2$) emissions from anthropogenic and, more recently, wildfire sources. However, most fire emission inventories do not include the reactive nitrogen compound nitrous acid (HONO), even though HONO was found to contribute at least 50% of a smoke plume's hydroxyl radicals (OH) for hours downwind of the fire (Peng et al., 2020; Theys

et al., 2020). HONO has the greatest relevance in the youngest parts of a smoke plume, where its rapid photooxidation (10–15 min lifetime at solar noon) generates OH, especially on plume edges (Decker et al., 2021; Palm et al., 2021; Wang et al., 2021). Thus, excluding HONO in fire emissions can greatly impact estimates of the initial chemical evolution of a smoke plume and generation of ozone and secondary organic aerosol (Wolfe et al., 2022).

After $NO_x$ is emitted, there are multiple pathways that can remove $NO_x$ from a smoke plume. $NO_2$ can react with oxidized acetaldehyde to form peroxyacetyl nitrate (PAN), a temporary reservoir species that breaks down in warmer temperatures and acts as a source of $NO_x$ downwind of a fire. $NO_x$ can also react with other radical species to form $RO_2NO_2$, $RONO_2$, $HNO_3$, $HO_2NO_2$, particulate nitrates, gas-phase organic nitrogen, nitrogen-containing VOCs and nitroaromatics. A number of studies have quantified the lifetime of $NO_x$ within wildfire smoke, under a variety of conditions, with estimates ranging

from 20 minutes to 11 hours (Adams et al., 2019; Akagi et al., 2012; Berezin et al., 2016; Griffin et al., 2021; Jin et al., 2021; Juncosa Calahorrano et al., 2021a; Takegawa et al., 2003; Wolfe et al., 2022).

In this study, we provide new remotely sensed measurements of vertical column HONO and $NO_2$ from the Sheridan Fire in the Prescott National Forest, Arizona taken roughly every 20 minutes over the course of the fire's activity on 16 August

2019. Additionally, we make estimates of the Sheridan Fire's emission rates and smoke plume effective lifetimes for every observation during its evolution. These measurements were made using the GeoCAPE Airborne Simulator (GCAS; Kowalewski and Janz, 2014) instrument aboard the National Aeronautics and Space Administration (NASA) ER-2 aircraft for the Fire Influence on Regional to Global Environments and Air Quality (FIREX-AQ; Warneke et al., 2023) campaign. We evaluate the assumptions inherent to the EMG approach to assess the EMG's utility at deriving emission rates and

lifetimes of wildfire smoke plumes. We use a simple one-dimensional (1-D) horizontal model to perform this evaluation. To improve upon the EMG approach, we provide a new emission rate and lifetime methodology using Monte Carlo 1-D model simulations and the diurnal FRP from geostationary satellites to derive emission rates and lifetimes for HONO and $NO_2$, using the Sheridan Fire as a test case.

## 2 Data and methods

## 2.1 FIREX-AQ

During the summer of 2019, a collaborative, multi-agency campaign called FIREX-AQ studied wildfires and agricultural fires in the continental United States to assess fires' impact on air quality and climate. One of FIREX-AQ's measurement

platforms was the NASA ER-2 high-altitude measurement aircraft. On the ER-2, the GCAS instrument remotely retrieved $NO_2$ and HONO vertical column densities. The GCAS instrument is composed of two push-broom spectrometers: the first records the spectrum as absolute nadir radiance in the ultraviolet to visible (UV-Vis), from 300 to 490 nm, and the second records the spectrum as absolute nadir radiance in the visible to near-infrared (Vis-NIR) from 480 to 900 nm (Kowalewski and Janz, 2014). The UV-Vis window has a spectral resolution of 0.6 nm with uncertainty in $NO_2$ slant column retrievals close to $0.8 \times 10^{15}$ molec $cm^{-2}$ (Judd et al., 2020). The GCAS instrument shares similar design specifications with the TROPOspheric Monitoring Instrument (TROPOMI; Veefkind et al., 2012) and Tropospheric Emissions: Monitoring of Pollution (TEMPO; Chance et al., 2013) instruments. TROPOMI operates in the 310 to 405 nm and 405 to 500 nm bands with a spectral resolution of 0.55 nm and in the 675 to 725 nm and 725 to 775 nm bands with a spectral resolution of 0.5 nm (Veefkind et al., 2012). TEMPO operates in the 290 to 490 nm and 540 to 740 nm bands with a spectral resolution of 0.57 nm (Zoogman et al., 2017). As flown on the ER-2 for FIREX-AQ, the GCAS data has a horizontal resolution of 500 m.

The ER-2 aircraft performed 10 flights between 2 August and 21 August 2019. In this work, we present observations from the Sheridan Fire. The Sheridan Fire was ignited by lightning on 5 August 2019 in Arizona's Prescott National Forest and continued to burn throughout the campaign. On 16 August 2019, the Sheridan Fire consumed 65% Pinyon-Utah juniper forest and 29% Turbinella oak-alderleaf mountain mahogany shrubland, where 80% of the total carbon emitted came from flaming conditions. On this same day, the ER-2 aircraft flew over the fire in such a way to create a sweeping bowtie pattern from an approximate altitude of 20.35 km, where the fire plume was captured on the downwind side of the bowtie and the background air was captured on the upwind side of the bowtie. This same fire was also sampled by the NASA DC-8 aircraft, equipped with several remote sensing and *in situ* observations of composition, near in time to the ER-2. The DC-8 flew above the Sheridan Fire smoke plume starting at approximately 24 UTC, or 17:00 local time, two hours after the ER-2 started to make measurements. This fire provides an ideal test case to compare plume composition determinations and constrain chemical models of plume evolution.

To approximate the winds driving the Sheridan Fire's smoke plume transport, we use reanalysis data, described in section 2.2. To estimate the height of the winds that we need to sample from the reanalysis data, we need to know the altitude of the smoke plume. The NASA Langley Research Center Differential Absorption Lidar-High Spectral Resolution Lidar (DIAL-HSRL) instrument on the DC-8 aircraft measured vertical profiles of aerosol backscatter at 532 nm, and we approximated the plume altitude to be the most concentrated part of the plume. Altitude was converted to a pressure level by recording the DC-8 aircraft data at a time when the DC-8 was flying through the smoke plume. The DC-8 plane's altitude and the plume's altitude were approximated to be at a pressure level of 588 mb.

## 2.2 GCAS Methodology

The retrieval method for GCAS measurements includes the differential optical absorption spectroscopy (DOAS) spectral fitting that yields slant column densities (SCDs) (Plane and Saiz-Lopez, 2006; Platt and Stutz, 2008). By fitting the differential cross-sections of trace gases to the differential absorption spectra, the trace gas concentrations along the light path can be determined via the Beer-Lambert law. The log-normalized GCAS spectra were fit to cross-section data using the software package QDOAS developed at the Belgian Institute for Space Aeronomy (BIRA-IASB) (Danckaert et al., 2012).

The fitting windows for $NO_2$ and HONO are 425 to 460 nm and 345 to 390 nm, respectively. The $NO_2$ fitting window is within the standard TROPOMI $NO_2$ product window of 405 to 465 nm (van Geffen et al., 2022). HONO absorbs in the UV-Vis spectrum at wavelengths 342 nm, 354 nm, and 368 nm (Stutz et al., 2000). As discussed in Lamsal et al. (2017), since the retrievals use average radiance from a clean background (reference location) due to the lack of solar irradiance measurements for normalization, the spectral fitting procedure provides differential slant column amounts which represent

slant columns with respect to the reference location.

Vertical column densities below aircraft are calculated using the differential SCDs and air mass factors (AMFs) following the approach discussed in Lamsal et al. (2017). An AMF is a quantity representing the effect that the light's path has on retrieval. An AMF depends on wavelength, altitude, observation geometry, surface reflectivity, a-priori vertical profiles, aerosols, and other factors that affect the measurement sensitivity. In this study, AMFs are calculated using the vector

linearized discrete ordinate radiative transfer code (VLIDORT), version 7.2 (Spurr, 2006). This radiative transfer code is a multiple-scattering model calculating radiances and weighting functions in a multilayer atmosphere. The AMFs for HONO and $NO_2$ are derived similarly to the procedure in Lamsal et al. (2017), where AMFs are calculated using non-Lambertian bidirectional reflectance distribution functions (Lamsal et al., 2017). A-priori profiles are taken from the NASA GEOS GMI

simulation at 0.25° x 0.25° resolution. Impact of aerosols is partially accounted for in the retrievals due to the use of average radiance measurements for the spectral fit as well as reference correction in the calculation of VCDs; retrievals are likely affected for high aerosol cases. Given the complex radiative transfer through evolving wildfire plumes, the VCD used here may carry significant uncertainties. However, our conclusions are focused mostly upon the comparisons of emissions and lifetime estimates, and less so on the absolute values of each trace gas. That said, we do provide comparisons of these

quantities to emission inventories and model estimates of chemical lifetime together with a discussion of the associated uncertainties in Sections 3.3 and 3.4.

## 2.3 ERA5

To estimate the wind speeds of the Sheridan Fire smoke plume, we obtained hourly reanalysis data from ECMWF Reanalysis v5 (ERA5), which has a horizontal resolution of 0.25° x 0.25° (C3S, 2018; Hersbach et al., 2023). We collected zonal and meridional wind speeds at pressures every 50 mb from 400 mb to 750 mb and every 25 mb from 750 mb to 1000 mb. With the smoke plume altitude information from the DIAL-HSRL, the wind speeds over the Sheridan Fire center were determined by interpolating the ERA5 zonal and meridional wind to the time that the ER-2 flew over the fire center, to the

longitude and latitude of the fire center, and to the pressure level of the fire plume.

## 2.4 GOES FRP

The Geostationary Operational Environmental Satellites-16 (GOES-16) (East) and GOES-17 (West) provide full-disk snapshots of Earth every five minutes. On both satellites, the Advanced Baseline Imager (ABI) uses visible and infrared spectral bands to locate fires and retrieve fire characteristics. FRP information was retrieved from both GOES-16 and GOES-

17 using the WildFire Automated Biomass Burning Algorithm from the University of Wisconsin, Madison (Schmidt, 2020). We acknowledge that the GOES-17 ABI has cooling system issues and thus impacts the FRP retrievals during this time period. We have opted to keep the GOES-17-derived FRPs to be consistent with previous FIREX-AQ analyses that use GOES-17 data (Peterson et al., 2022; Warneke et al., 2023; Wiggins et al., 2020, 2021). A diurnal profile of the Sheridan Fire's sum-FRP, a sum of all fire pixel FRPs associated with the Sheridan Fire, was generated by selecting GOES data

within 4 km of the final fire perimeter from the Geospatial Multi-Agency Coordination (GeoMAC). GeoMAC is an internet-based mapping tool that stores wildfire perimeters in the contiguous 48 states of the United States and Alaska since 2000. A pseudo-diurnal FRP product was created to represent a more realistic diurnal fire cycle by reallocating 5% of the total daily FRP to the quiescent FRP periods where there are no FRP observations. We smoothed the transition between the quiescent and active FRP periods.

## 180 2.5 Emission inventories

The derived emission rates from this work are compared to a set of commonly used biomass burning emission inventories, including GFED4s (Giglio et al., 2013; van der Werf et al., 2017), FINNv2.5 (Wiedinmyer et al., 2023), QFEDv2.5 (Darmenov and da Silva, 2015), and GFASv1.2 (Kaiser et al., 2012). GFED4s is burned area emissions inventory that includes small fires. Monthly emission estimates are calculated by combining burned area maps from 500 m Moderate

Resolution Imaging Spectroradiometer (MODIS) data with the Carnegie-Ames-Stanford Approach model, which estimates fuel loads and combustion completeness (Potter et al., 1993). Daily emissions estimates stemmed from MODIS data. Conversion to emission rates is achieved by multiplying emission factors to the computed dry matter emissions (Akagi et al., 2011). The spatial resolution is $0.25° \times 0.25°$ and provides emission data for $NO_x$ (as NO). Emissions data is not provided for $NO_2$ or HONO.


FINNv2.5 is another burned area emissions inventory. It uses the 375 m Visible Infrared Imaging Radiometer Suite (VIIRS) and MODIS at 1 km$^2$ resolution to report active fire detections from which burned area is derived. Emissions are calculated from the following equation, Eq. (1):

$$E_i = A \times B \times FB \times EF_i,$$ (1)

where $i$ is a specific compound, $E$ is the emissions (g), $A$ is the area burned (m$^2$), $B$ is the amount of biomass (kg m$^{-2}$), $FB$ is the fraction of biomass burned (unitless), and $EF$ is the emission factor with units of mass of $i$ per mass biomass burned (g kg$^{-1}$). The emission factors used in FINNv2.5 are based on updates in the published literature (Akagi et al., 2011; Fang et al., 2017; Liu et al., 2016, 2017; Paton-Walsh et al., 2014; Santiago-De La Rosa et al., 2018; Stockwell et al., 2015; Urbanski, 2014). NO$_2$ is a standard emission product in FINNv2.5. FINNv2.5 can also derive other emission products from total non-

methane organic gases (NMOG) using three commonly used chemical mechanisms: Statewide Air Pollution Research Center Mechanism (SAPRC99; Carter, 1999), Model for Ozone and Related chemical Tracers (MOZART; Emmons et al., 2020), and Goddard Earth Observing System with Chemistry (GOES-Chem; Bey et al., 2001). FINNv2.5 can derive HONO from SAPRC and MOZART, but not from GOES-Chem. The spatial resolution of this inventory is 0.1° × 0.1° and the temporal resolution is daily.


QFEDv2.5 is an FRP-derived emissions inventory. FRP is attained from the MODIS Level 2 products, MODIS/Terra Thermal Anomalies/Fire (MOD14) and MODIS/Aqua Thermal Anomalies/Fire (MYD14). An emissions rate is calculated from the following equation, Eq. (2):

$$E_i = \alpha \times EF_i \times \frac{FRP}{A},$$ (2)

where $E_i$ is the emission rate of compound $i$ per unit area (g m$^{-2}$ s$^{-1}$), $\alpha$ is a constant that relates time integrated FRP (fire radiative energy) to dry biomass burned (kg J$^{-1}$), $EF_i$ is the emission factor of compound $i$ (g kg$^{-1}$), and $A$ is the area of the satellite pixel (m$^2$). The emission factors used in QFEDv2.5 are defined by those in Andreae and Merlet (2001). The spatial resolution is 0.1° × 0.1° and provides daily mean emission data for NO, but not NO$_2$ or HONO.

GFASv1.2 is also an FRP-derived emissions inventory using the MOD14 and MYD14 products from the MODIS instruments on the Terra and Aqua satellites, respectively. A dry matter combustion rate is calculated by multiplying land-cover-dependent conversion factors with the FRP areal density. The conversion factor relates FRP to dry matter burned and FRP areal density is total FRP in a grid cell divided by total observed area in a grid cell. Compound emission rates are found by multiplying an emission factor by the dry matter combustion rate. Emission factors are defined by those in Andreae and

Merlet (2001). The spatial resolution is 0.1° × 0.1° and provides daily mean emission data for NO$_x$ as NO, but not NO$_2$ or HONO.

## 2.6 Analysis methods

### 2.6.1 Calculation of HONO and NO₂ line densities

We transformed the ER-2 GCAS HONO and $NO_2$ VCD data into line densities by the following procedure. First, we
regridded each GCAS swath within each flight track onto grids with a resolution of 0.0045° (0.5 km), the resolution of the
instrument, and interpolated across portions of the grid with missing data. We then subtracted from the entire scene the
average HONO and $NO_2$ VCDs upwind of the fire, called the background HONO and $NO_2$ VCDs, resulting in enhanced
HONO and $NO_2$ VCDs solely from the wildfire. This process also removes any stratospheric component of HONO and $NO_2$.
Subsequently, we rotated the grids by the angle of the $NO_2$ VCD plume averaged over time, acting as a proxy of the mean
wind direction (Fig. A1). In general, the plume direction may not always align with the wind direction. We applied this
rotation methodology to the Sheridan Fire because of its near-ideal plume characteristics, a linear plume shape. Finally, we
summed along the plume in the crosswind direction to produce line densities. This process is visually summarized in Fig. 1.
For use with an hourly space-based instrument like TEMPO, only the regridding resolution would differ in this procedure
(0.02° for TEMPO). This lower resolution in grid size results in a lower resolution in line density, where the fine structures
seen with the GCAS instrument will not be as resolved.

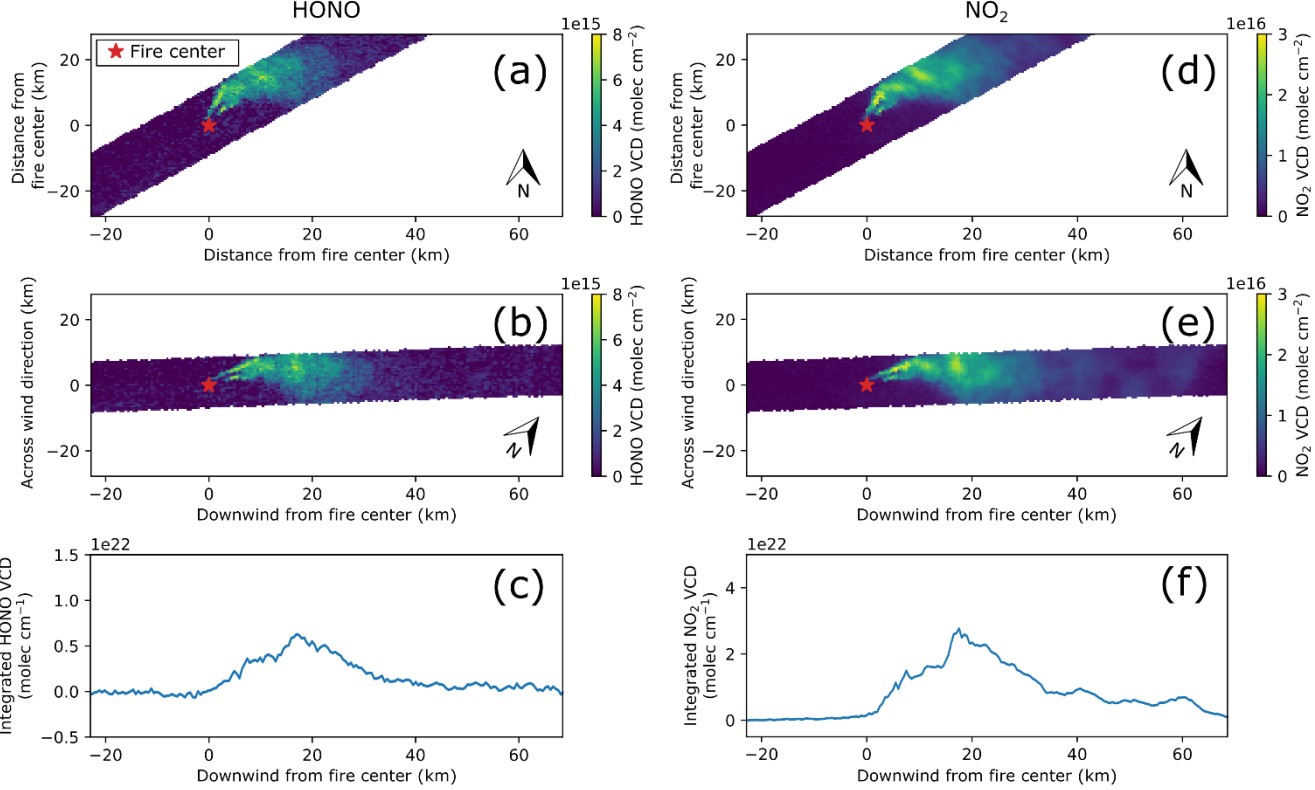

**Figure 1: Image of (a) gridded and (b) rotated track 14 GCAS HONO VCD. (c) HONO line density from track 14. (d)-(f) Similar to (a)-(c), but for NO₂.**

The resolution of the HONO image is lower than the NO$_2$ image because the magnitude of NO$_2$ VCD is approximately four times greater than HONO. This results in a higher signal-to-noise ratio for NO$_2$. Due to the sweeping bowtie flight pattern, only flight tracks 12, 14, and 18 capture the smoke plume core by sampling nearly parallel to the wind direction. Other tracks capture the smoke plume at an angle, thus line densities at these times will be incomplete and asymmetric. For the rest of this manuscript, emissions rates and lifetimes will be calculated solely for tracks 12, 14, and 18.

### 2.6.2 Exponentially Modified Gaussian (EMG)

In addition to extrapolating from burned area or FRP observations, emissions from biomass burning can also be estimated by directly analyzing the smoke plume chemical concentrations and shape. One such approach relies on fitting an EMG to the line density of, for example, daily satellite observations of NO$_2$ columns from TROPOMI (Jin et al., 2021). The EMG probability density function is the result of a convolution of the exponential and normal probability density functions. EMGs have been applied to satellite observations of the Ozone Monitoring Instrument (OMI) and TROPOMI NO$_2$ to estimate emissions and lifetimes from point sources and urban areas (Beirle et al., 2011; De Foy et al., 2015; Goldberg et al., 2019; Laughner and Cohen, 2019; Lu et al., 2015; Xue et al., 2022). Two-dimensional EMGs have been used to estimate SO$_2$ and NH$_3$ emissions and lifetimes (Dammers et al., 2019; Fioletov et al., 2015; McLinden et al., 2020). In addition to satellite data, EMGs were used to estimate emissions of NO$_x$ and NO$_2$ from wildfires with data from field campaigns (Griffin et al., 2021).

The EMG function we use in this study is modeled after Jin et al. 2021 and is defined in Eq. (3):

$$L(x|a, x_0, \mu_x, \sigma_x, B) = \frac{a}{2x_0} \exp\left(\frac{\mu_x}{x_0} + \frac{\sigma_x{}^2}{2x_0{}^2} - \frac{x}{x_0}\right) \text{erfc}\left(-\frac{1}{\sqrt{2}}\left(\frac{x - \mu_x}{\sigma_x} - \frac{\sigma_x}{x_0}\right)\right) + B, \tag{3}$$

where $a$ is a scale factor representing the total number of molecules in a plume, $x_0$ is the $e$-folding distance representing the length scale of the exponential decay in km, $\mu_x$ is the location of the apparent source relative to the source center in km and is the center of the Gaussian component, $\sigma_x$ is the square root of the variance of the Gaussian component in km, $B$ is the background in molec km$^{-1}$, and erfc is the complementary error function (Jin et al., 2021). Best guesses for initial values were made following Laughner and Cohen (2019) but in the event that a fitting failed, parameters were manually nudged until the fitting function settled on a solution that mimicked the sample data (Laughner and Cohen, 2019).

From the EMG parameters, an emission rate ($E_{EMG}$; molec s$^{-1}$) and effective lifetime ($\tau_{EMG}$; s) can be estimated with Eqs. (4) and (5):

$$\tau_{EMG} = \frac{x_0}{w}, \tag{4}$$

$$E_{EMG} = \frac{a}{\tau_{EMG}}, \tag{5}$$

where $w$ is the wind speed. The effective lifetime closely represents a chemical lifetime if the emissions, wind speed and direction are constant, and no deposition occurs (De Foy et al., 2014). The wind speeds used in our EMG fits are the same as described in section 2.2. We will show how the wildly varying temporal behavior of wildfires challenges the basic application of this method.

### 2.6.3 PECANS model

The Python Editable Chemical Atmospheric Numeric Solver (PECANS) model is a flexible, idealized atmospheric chemistry multi-box plume model with Gaussian emissions and idealized transport (Joshua-Laughner and Laughner, 2023; Laughner and Cohen, 2019). In version 0.1.1, users specify the dimensionality of the model, choose to include first-order chemistry and emissions, set an initial chemical condition, and set a constant emission rate.

For this research, we ran 1-D PECANS simulations in four configurations that vary the shape of the emission rates with time: constant emissions, step-change emissions, Gaussian emissions, and FRP-profile emissions. In the first configuration, the emissions rate is kept constant for the entire modeled run time and is the simplest model configuration. In the second configuration, the emissions rate has a step-change halfway through the modeled run time, adding complexity to the modeled emissions rate. In the third configuration, the emissions rate is multiplied by the probability density function of a Gaussian with a mean of 5,000 s and standard deviation of 1,000 s, as daily fire activity has been modeled with a Gaussian distribution previously (Andela et al., 2015). Finally, in the fourth configuration, the emissions rate is prescribed such that for every time step, the total daily emission rate is multiplied by the fractional FRP, representing a data assimilation scenario since emission rates have been modeled as functions of FRP. Fractional FRP is calculated by first interpolating the 5-min pseudo-diurnal FRP product to every second, then by normalizing with the daily total FRP. The last three configurations require manual edits to the emissions_setup.py file in the PECANS code, where multiplication factors are applied to the emissions time series. In a limited run of sensitivity tests, we found that the diffusion coefficients in the x and y dimensions did not significantly affect the EMG calculation of emission rate and lifetimes. A more detailed description of the base model parameters is in Table B1.

## 3 Results and discussion

### 3.1 HONO and NO$_2$ plume structure in the Sheridan Fire

In multiple flight overpasses, the GCAS on the ER-2 captured the structures of HONO and NO$_2$ plumes evolving over time. In Fig. 2, we show the rotated, gridded GCAS VCDs for HONO and NO$_2$ at three moments in time: ER-2 track 12, track 14, and track 18. In track 12, there are two local maxima of HONO and NO$_2$. The first local maximum is centered about 7.5 km downwind of the Sheridan Fire for both HONO and NO$_2$. The second local maximum is centered around 12.5 km downwind for both compounds. HONO and NO$_2$ plumes share the same plume edges, but the NO$_2$ plume measurement signal persists through 50 km downwind from the fire, while the HONO signal is lost to noise and the plume edge definition vanishes. In track 14, both HONO and NO$_2$ show two parallel plume lines originating from the fire center, as well as another local maximum around 18 km downwind of the fire. Both plumes have lower VCDs than those detected in track 12. Finally, by the time track 18 occurs, the HONO signal is barely present, with the instrument detecting HONO only as far as 20 km downwind of the fire. However, enhanced NO$_2$ remains detectable as far as 60 km away from the fire. Overall, in all three overpasses, HONO and NO$_2$ share local maxima, plume edges, and plume shape. This means that both chemicals shared similar emission characteristics, were exposed to a consistent dispersal pattern, and traveled together under the same atmospheric conditions.

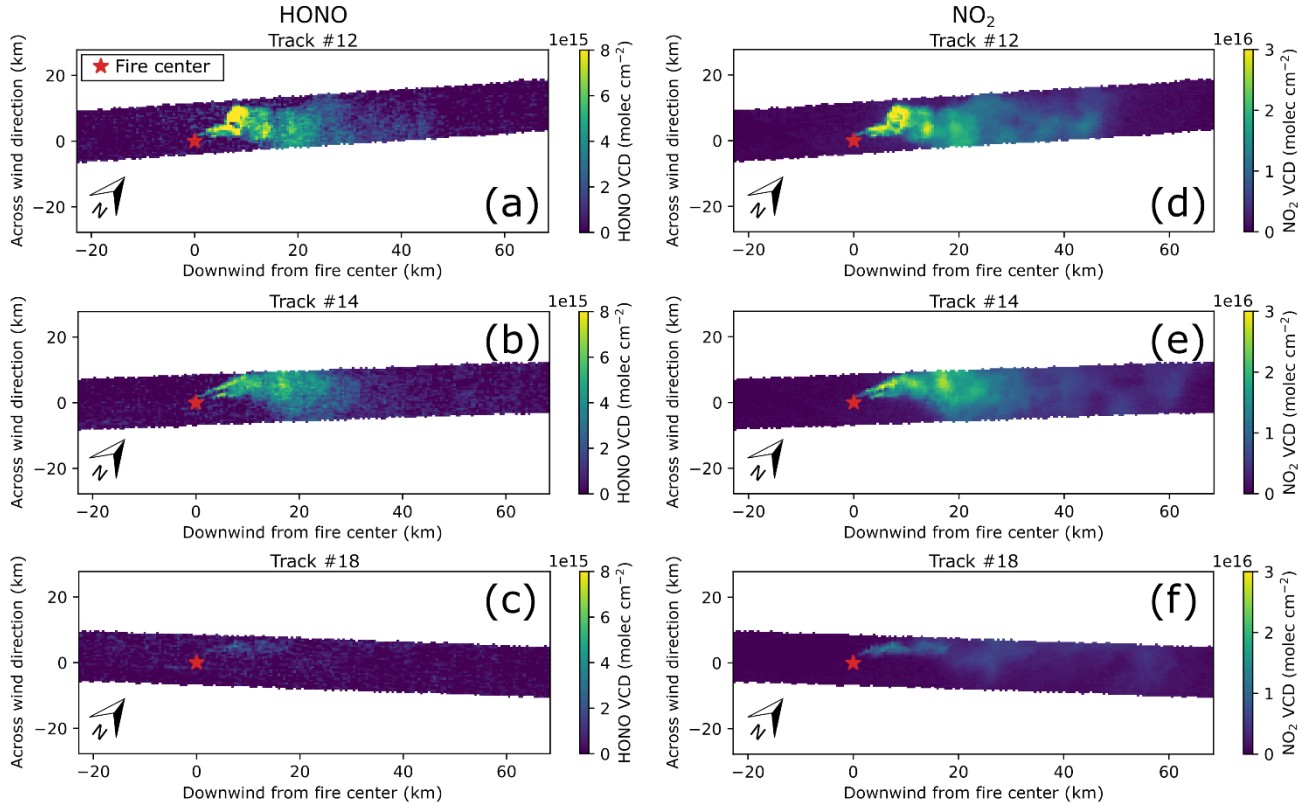

**Figure 2: Images of rotated GCAS HONO VCD for tracks (a) 12, (b) 14, and (c) 18. Images of rotated GCAS NO₂ VCD for tracks (d) 12, (e) 14, and (f) 18.**

### 3.2 EMG emission rates and lifetimes from the Sheridan Fire

Emissions from the Sheridan Fire, using sum FRP as a proxy, varied widely over the course of four hours on 16 August 2019 (Fig. 3a). There are features in the GCAS line densities that clearly deviate from an EMG shape. In Fig. 3e, track 12 appears to decrease sharply in integrated $NO_2$ VCD around 25 km away from the fire center, and another that occurs around 50 km away from the fire center. This change leads to the EMG fit underestimating the line density between 25 and 50 km and overestimating the line density from 50 km to the end. Given that the wind speed is approximately 10 m s$^{-1}$, we infer that the edge of the plume is at 50 km and thus the steep decay is not related to a chemical decay, but a physical edge. HONO appears to have the sharp decline in integrated HONO VCD that $NO_2$ demonstrates in track 12 (Fig. 3b), but due to the higher uncertainty in retrievals and its shorter lifetime via photolysis, higher noise in HONO's line density may hide spikes that were present with $NO_2$. This shorter lifetime of HONO may enable use of the standard EMG fit even with time-varying emission rates.

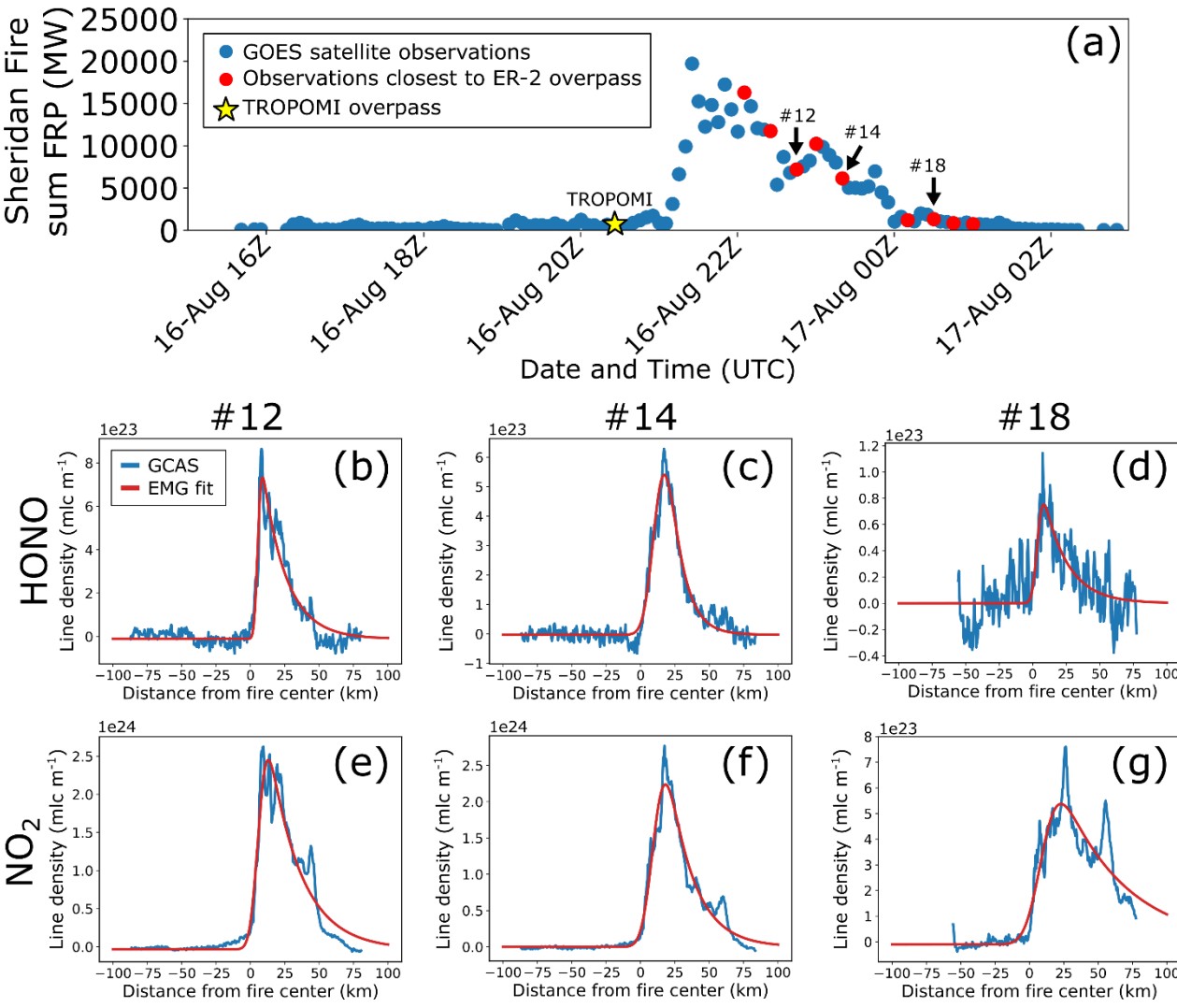

**Figure 3: (a)** Sum GOES-16 and GOES-17 FRP in blue with ER-2 overpass times in red. Arrows link ER-2 overpass track numbers to the plots below and the TROPOMI overpass time is labeled with a yellow star. **(b)** Track 12, **(c)** track 14, and **(d)** track 18 ER-2 GCAS HONO line densities (blue) fit to an EMG (red). **(e)** Track 12, **(f)** track 14, and **(g)** track 18 ER-2 GCAS NO₂ line densities (blue) fit to an EMG (red). Note that y-axis limits are adjusted for each track's maximum HONO and NO₂ value.

In Figs. 3c and 3f, the track 14 line densities, the EMG fit peaks do not capture the observational maximums, but the EMG fit better captures the observational tails versus their counterparts from track 12 possibly because the smoke plume being further transported and processed is closer to its inherent steady-state plume shape. Before analyzing the track 18 line densities, it is crucial to note that sum FRP at track 18 is back to the background value. The $NO_2$ line density maximum for track 18 in Fig. 3g is nearly four times smaller than that of track 14. We assume that the emission rates and lifetimes of

HONO and NO₂ for track 18 are inaccurate because the EMG fit cannot capture the fact that the fire subsided to near zero power, thus emissions should be zero. This analysis demonstrates that the EMG functional fit approach for estimating emission rate and lifetime is not always appropriate within a fire's diurnal cycle: track 12 did not have enough time to reach steady state, track 18 was sampled after the fire died down for the day, and track 14 was between these extremes. It would not be possible to assess EMG fits in this way applied to single (once a day) overpasses of a fire.

Acknowledging that HONO and NO₂ have the same sampling biases, meaning that each GCAS sample of the plume shares the same sampling orientation of the sweeping bowtie, we can take the ratio of the emissions rates of HONO and NO₂ ($E_{EMG,HONO}/E_{EMG,NO2}$) from all flight tracks to explore how reactive nitrogen in wildfire smoke is partitioned and processed. By taking a ratio, the sampling biases in HONO and NO₂ VCDs and EMG emission rates cancel out. In Fig. 4a, the HONO/NO₂ emission ratio decreases with time. A decrease in the emission ratio indicates that over time, there is a relative decrease in HONO in reference to previous values. Additionally, we find that the HONO/NO₂ emission ratio increases with sum FRP (Fig. 4b). This contradicts the foundations of biomass burning emission inventories, which rely on constant emission factors that only vary by land cover type. If this were true, HONO and NO₂ should have constant emission ratios over both time and sum FRP. However, variations in the emission ratio span a factor of 3 over the course of 2.5 hours. This FRP-dependent behavior of HONO relative to NO₂ was reported previously based on satellite-retrieved concentrations closest to Australian wildfires (Fredrickson et al., 2023).

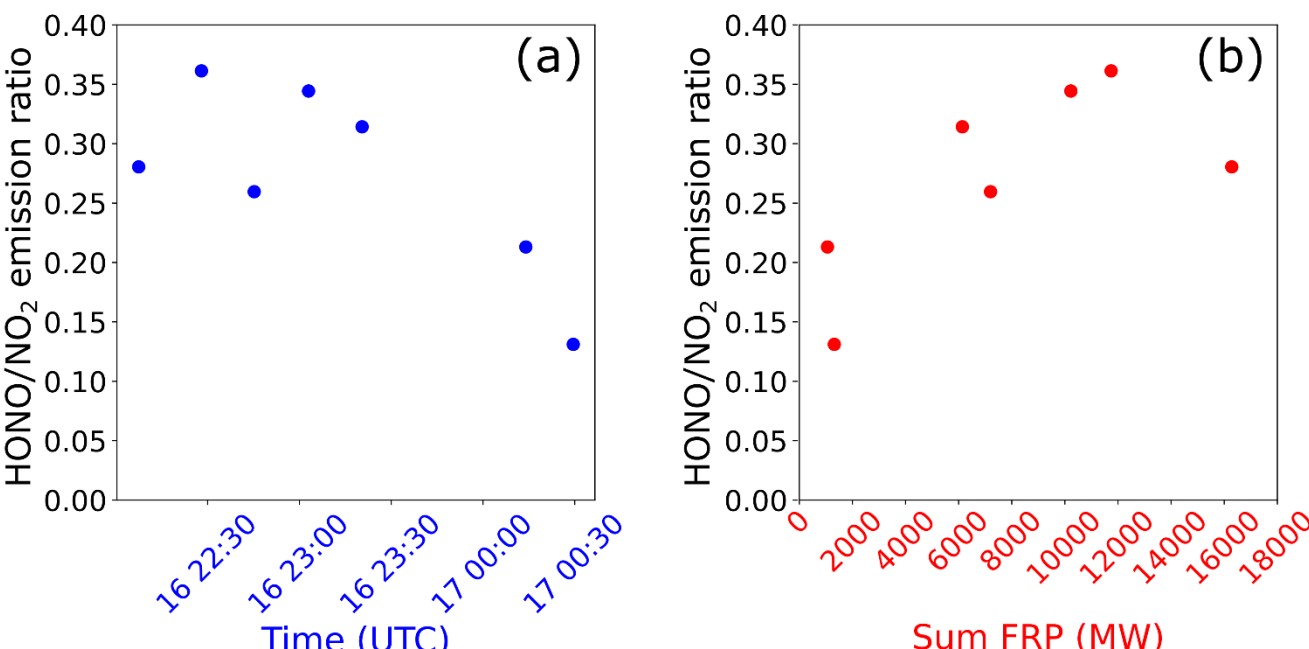

Figure 4: The emission ratio of HONO to NO₂ plotted against (a) time and (b) sum FRP for the Sheridan Fire.

### 3.3 An improved EMG methodology: Monte Carlo diurnal 1-D models

As alluded to previously, a basic application of the EMG method is rarely suitable for daily observations of wildfires as applied in previous studies for a few reasons (Griffin et al., 2021; Jin et al., 2021). First, a fire may not have had enough time to have its smoke and emission products transported to provide a complete picture of the decay due to loss processes. The EMG fit would therefore underestimate the lifetime. We found evidence of this behavior in Fig. 3e. Additionally, if a fire subsides and dies out, but the plume is still being transported, the EMG fit would still assign a substantial emission rate to

the fire. The EMG method in this situation disconnects the fire from the plume, as seen in Fig. 3g. These situations are complex and nonideal, and the EMG method is not suited for them. This then begs the questions of what situations the EMG method is suited for; how long a fire needs to burn before the EMG method provides an accurate result; how quickly the EMG method responds to a change in fire emissions; and how the EMG method stands against a simplified diurnal emission profile.

In Fig. 5, we ran idealized simulations of a hypothetical compound to analyze the accuracy of EMG fits over time. As shown in Figs. 5a and 5b, the smoke plume from an idealized 1-D simulation with constant emissions (see Section 2.5.3 and Table B1 for model configuration) reaches steady state within the modeled run time. While the model emissions source is centered at 25 km, the line density peak is shifted to the right due to wind transport. We fit an EMG to the model output every 200 s

modeled run time and compared the fit results to the model inputs by plotting the ratio of the fit parameter to the model prescribed parameter, shown in Fig. 5c. At modeled run times greater than 5000 s, the ratios of EMG fit parameters and model prescribed values for both the emission rate and lifetime approach 1, indicating near equivalence and thus accurate EMG results. However, when fitting EMGs to plumes modeled at earlier times, the lifetime is underestimated by as much as 97% while the emission rate is overestimated up to 550%.

Second, wildfire emissions and intensity change over time on an hourly or shorter timescale, while the EMG profile shape assumes that the emission rate has been constant. In Figs. 5d and 5e, the emission rate used in the model is halved halfway through the simulation, i.e., the step-change emissions simulation. The steady-state maximum in the line density near 25 km starts to decrease after emissions are halved and a secondary peak in line density is observed traveling to the right as the

385 initial high-emission plume decays. After the emissions are halved, comparison with the EMG fit/model prescribed ratios show that the emission rate is at first overestimated, then slightly underestimated (Fig. 5f). On the other hand, the lifetime is at first unaffected by the change in emission rate, but then becomes overestimated as the peak from the high-emission times is transported downwind and lengthens the line density shape.

In Figs. 5g and 5h, we applied the EMG method to an idealized simulation with temporally varying emissions that followed a Gaussian, which is a better approximation of fire behavior than the previous simulations. Instead of the presence of two line density peaks as in Fig. 5e, only one line density peak persists through the model simulation (Fig. 5h). By the end of the model run, the line density peak has shifted from roughly 25 km to 70 km downwind. As shown in Fig. 5i, the EMG determined emission rate initially overestimates the model prescribed value, but then approaches the true value from 1000 to 2500 s. As the emission rate grows (Fig. 5g) with time, the EMG determined emission rate separates from the model prescribed value, overestimating once more, and eventually grows towards infinity (Fig. 5i). This behavior occurs because the Gaussian emission profile approaches zero near the end of the model run, while the plume persists. The EMG determined lifetime, on the other hand, is consistently underestimated. This behavior occurs because not enough time has passed for the inherent line density shape to be realized before the modeled emission rate starts to decline after reaching peak emissions. This decrease in emission rate leads to the shifting of the line density peak downwind, shortening the e-folding distance used to calculate the lifetime. Thus, shifts in emissions will lead to shifts in lifetime, making this Gaussian simulation even more complicated for the EMG method to fit.

Third, wildfires have been observed to increase their thermal output after the 1:30 PM local time that TROPOMI and some other polar-orbiting satellites observe at (Wiggins et al., 2020), and thus daily EMG estimates from polar-orbiting satellites may underestimate maximum emission rate. We see that for the Sheridan Fire, TROPOMI missed the fire altogether, as TROPOMI made its overpass around 20:25 UTC (Fig. 3a).

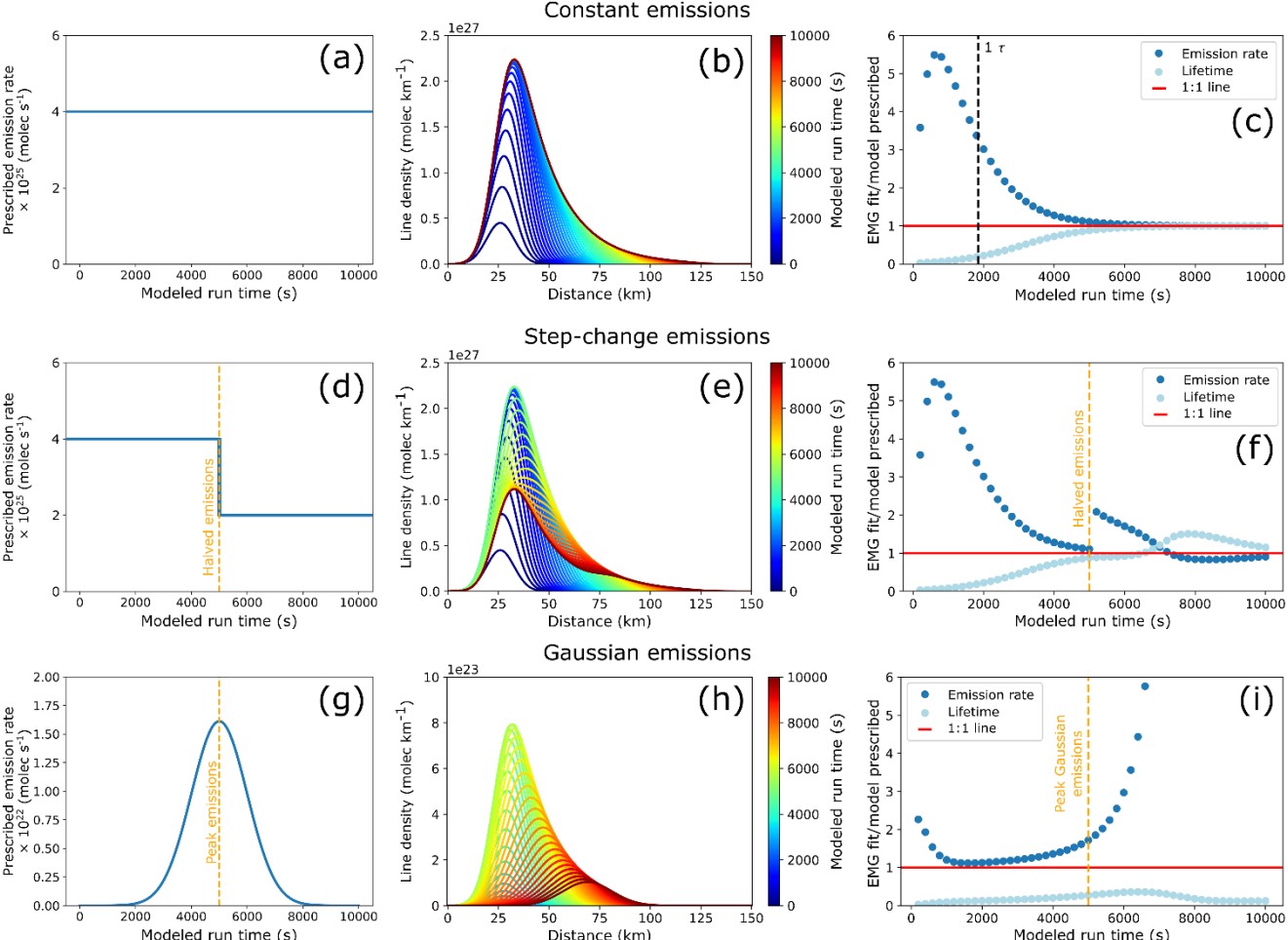

Figure 5: (a) The emission profile used as input to the idealized 1-D constant emissions model. (b) Line densities of a hypothetical compound from the 1-D model with prescribed chemical lifetime of 30.8 minutes (black, dashed line) and constant emissions colored by modeled run time. (c) EMG fit / model prescribed emission rate (blue) and lifetime (light blue) at every model output step through the course of the entire modeled run time. The modeled time equal to the model lifetime is indicated with a vertical, black dashed line. (d) The emission profile used as input to the idealized 1-D step-change emissions model. (e) Line densities from the 1-D model where the emission rate is halved halfway through the modeled run time, colored by modeled run time. (f) Similar to (c), but for the model in (e). The vertical, gold dashed line indicates where the emissions were halved. (g) The emission profile used as input to the idealized 1-D Gaussian emissions model. (h) Line densities from the 1-D model where the emissions are scaled to a Gaussian PDF with a mean of 5000 and standard deviation of 1000, colored by modeled run time. (i) Similar to (c), but for the model in (h). The vertical, gold dashed line indicates peak Gaussian emissions.

The above results illustrate that the application of EMGs to wildfire plumes should improve with hourly observations of wildfire pollutants as will become possible with the TEMPO instrument or another geostationary satellite instrument that measures atmospheric composition. As emission rates are initially overestimated from the EMG approach early in the fire growth, using observations that are at least forty minutes later than fire start (past some FRP threshold; see Fig. C1), and

425 analogously discarding observations after the fire intensity has diminished will both increase accuracy of the EMG estimates of emissions and lifetime.

The EMG method is inherently limited in that the emissions and lifetime are coupled due to an underdetermined system. We address this issue by improving constraints on the time-varying emissions using observed FRP variations together with a
430 Monte Carlo application of the 1-D model to sample a large parameter space of lifetime and emission conditions. First, the diurnal FRP shape is obtained by dividing the summed GOES-16 and GOES-17 5-minute FRP product by the GOES total daily FRP for times 16 August 7 UTC to 17 August 7 UTC, equivalent to 16 August 00:00 to 23:59 local time. Second, EMG functions are fit to all appropriate GCAS observations (greater than forty minutes after fire start and before fire intensity diminishes) to create a population of EMG fit emission rates and lifetimes and establish sampling ranges. Third, the
435 ranges of EMG fit emission rate divided by the fractional FRP (total daily emission rate) and lifetime are used to create sampling distributions for a Monte Carlo simulation. We run this fourth configuration of the 1-D PECANS model one hundred times, varying the lifetime and total daily emission rate parameters based on the distributions. The root mean square error (RMSE) between the model and the GCAS observations for tracks 12 and 14 are shown in Fig. 6 for HONO and $NO_2$. The absolute minimum RMSE provides an indication of the most likely lifetime and emissions rate combination(s).

In Fig. 6, RMSE is approximately constant in a combination of lifetime and emission rate values that follow an inverse relationship, which is expected since general mass balance constitutes that concentration is proportional to emissions and inversely proportional to loss rate. In Figs. 6a and 6b, the smallest RMSE between the HONO observed and model derived line densities occur with a lifetime between 15 and 27 minutes and an emissions rate of 0.6 to $1.75 \times 10^{25}$ molec s$^{-1}$. In Figs.
6c and 6d, $NO_2$ has a larger viable range of lifetimes and total emissions rates, which range between 20 and 45 minutes and 2.0 to $5.25 \times 10^{25}$ molec s$^{-1}$, respectively. Track 18 was excluded from this analysis as measurements occurred after the Sheridan Fire had ceased its high emissions for the day. The model runs with the smallest RMSE values are shown in Fig. D1. In Fig. D1d, the model enables EMG fits to non-EMG shapes in the observed multi-model line density, which is not possible with the basic application of the EMG method (Fig. 3f).

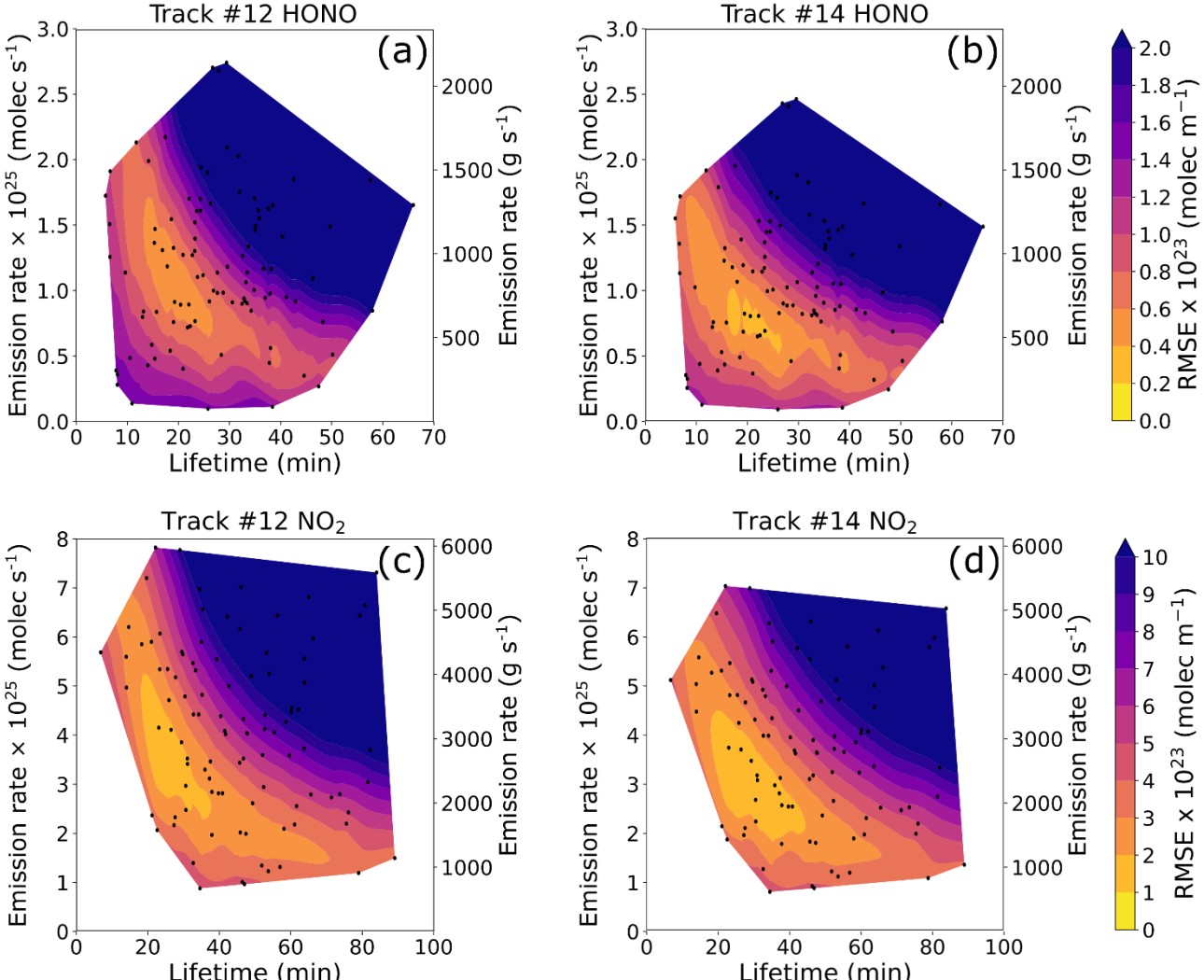

**Figure 6: The RMSE value between 100 PECANS models with varied total emission rate and lifetime model input parameters and the observed GCAS HONO VCD line density for (a) track 12 and (b) track 14. The emission rate is plotted by multiplying the total emission rate by the normalized FRP at the time of each track. A filled contour map creates a 2D interpolation of the randomly sampled models (dots). (c) and (d) Similar to (a) and (b), but for GCAS NO₂ VCD.**

The lifetime of HONO from photolysis alone just above the smoke plume, estimated from the TUV Quick Calculator (Madronich, 2016), would be 11 min and 13 min during the overpasses of track 12 and 14 respectively, which is shorter than those lifetimes found in Figs. 6a and 6b. The inputs and outputs of the TUV Quick Calculator can be found in Tables E1 and E2. Likely, HONO is photolyzing on the plume edges, but HONO deeper in the plume likely has longer lifetimes as proposed in earlier literature (Wang et al., 2021). The $NO_2$ lifetime for the Sheridan Fire we find from the Monte Carlo approach is smaller than most cited in the literature and still smaller than those values used in model assumptions. In

multiple satellite $NO_x$ observational studies, a $NO_x$ lifetime of 2 hours is typically assumed to infer mass emission rates and emission coefficients of $NO_x$ from wildfires, citing previous field and model studies (Mebust et al., 2011; Mebust and Cohen, 2014). Other studies use a $NO_x$ lifetime of 6 hours for fires, citing lifetimes of $NO_x$ found using satellite observations over megacities, as well as from an aircraft campaign sampling biomass burning plumes in northern Australia (Schreier et al., 2014, 2015; Tanimoto et al., 2015). Other studies report $NO_x$ lifetimes within or above this range (Akagi et al., 2012; Berezin et al., 2016; Takegawa et al., 2003). Only two studies reported $NO_x$ lifetimes under 2 hours in wildfire plumes sampled with aircraft in the western United States (Juncosa Calahorrano et al., 2021b; Wolfe et al., 2022). One other satellite study used OMI $NO_2$ satellite retrievals from the 2016 Horse River wildfire in Alberta Canada and reported a range of $NO_2$ lifetimes from 1 to 2.5 hours (Adams et al., 2019). Two other studies used EMG and flux methods to estimate the lifetime of $NO_x$ from wildfires. Jin et al. (2021) reported a range of $NO_x$ lifetimes from 0.8 to 10.5 hours using the EMG method and Griffin et al. (2021) reported three different ranges based on the method (Griffin et al., 2021; Jin et al., 2021). Using EMG on TROPOMI observations, the $NO_x$ lifetime ranged from 1 to 3 hours, while EMG applied to field campaign data ranged from 0.9 to 6.5 hours and using EMG on model VCDs ranged from 0.5 to 1.5 hours (Griffin et al., 2021).

It is beyond the scope of this paper to provide an in-depth exploration of the source of our shortened $NO_2$ lifetime, especially given the single fire we examine here. It is worth repeating that the EMG method derives an effective lifetime, not a chemical lifetime. The effective lifetime also includes influences from plume meandering, deposition, sampling issues, and other factors, meaning that effective lifetimes are usually shorter than chemical lifetimes (De Foy et al., 2014; Lu et al., 2015). However, a simple chemical box model applied to the Sheridan Fire indicates that a large source of organic peroxy radicals ($RO_2$) is needed to drive the $NO_2$ chemical lifetime to the short values we infer from applying the 1-D plume model (with no chemical mechanism) to the GCAS observations. This is not dissimilar to a hypothesized missing $RO_2$ source seen in the Taylor Creek Fire from 2018 (Peng et al., 2021). In that study, a baseline box model simulation of the Taylor Creek Fire overestimated $NO_x$ and underestimated organic nitrates, therefore missing a $NO_x$ to organic nitrate reaction pathway. By incorporating proxy peroxy radicals with organic nitrate formation pathways, their model better reflected the observed $NO_x$ decay. This has crucial implications for understanding the chemical evolution of wildfire smoke.

### 3.4 Biomass burning emission inventories underestimate observationally constrained emission estimates

We compare our Monte Carlo diurnal 1-D model HONO and $NO_2$ emissions with those reported in other biomass burning emissions inventories. We first convert the total emission rate from the lowest RMSE diurnal 1-D model to a daily mean emission rate. The total emission rate is multiplied by the normalized diurnal FRP profile to get a diurnal profile of instantaneous emission rates. This diurnal profile is then averaged to produce the daily mean emission rate. As shown in Fig. 7a, FINNv2.5 HONO emissions for the SAPRC and MOZART mechanisms underestimate the track 12 and track 14 daily mean emission rate of our method (MC diurnal 1-D) by a factor of 9.4 for track 12 and a factor of 7.2 for track 14. In

contrast our Monte Carlo diurnal 1-D model $NO_2$ emissions estimates fall in the middle of the four emissions inventories. Our method exceeds the estimates of bottom-up approaches but is exceeded by the estimates of top-down approaches. QFED has the largest $NO_2$ emissions and is nearly 5 times the GFASv1.2 estimate. For the Sheridan Fire, HONO emissions are underestimated by biomass burning emission inventories, but $NO_2$ emissions are in the range estimated by the other inventories.

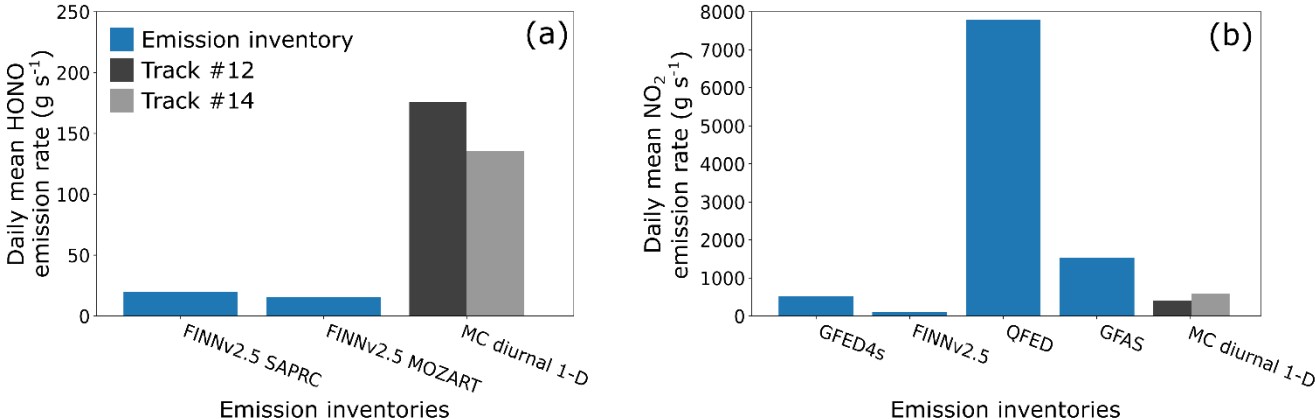

**Figure 7: Daily mean emissions of (a) HONO and (b) $NO_2$ from the Monte Carlo diurnal 1-D model method (MC diurnal 1-D) using the lowest RMSE model for each track to define the emissions, and the following biomass burning emission inventories: FINNv2.5 SAPRC, FINNv2.5 MOZART, GFED4s, FINNv2.5, QFEDv2.5, and GFASv1.2.**

As discussed earlier, the VCD retrievals from the GCAS instrument are subject to uncertainties, primarily uncertainties in the AMF determination. One major source of AMF uncertainty is the presence of and the characteristics of aerosols in the retrieval columns (Cooper et al., 2019). Aerosols can increase the AMF through scattering; scattering can increase the light path or increase the radiance observed by remote sensing instruments. Aerosols can also decrease the AMF by shielding a compound below a layer from remote sensing instruments or absorbing aerosols can reduce the scattering back towards remote sensing instruments. Wildfire smoke is made up of both scattering and absorbing aerosols and can be present as dilute or concentrated plumes. With both HONO and $NO_2$ enhancements coinciding within the aerosol layer, we hypothesize that aerosols both shield HONO and $NO_2$ deeper within the plume and absorb sunlight. This will likely lead to an underestimation of HONO and $NO_2$ concentrations and therefore an underestimation of emission rates. Thus, the actual emissions may be more on par with the QFED and GFAS inventories.

# 4 Conclusions

Using high-resolution remote observations of fire plumes from the GCAS instrument on the NASA ER-2 during the FIREX-AQ campaign, we estimated the evolving emission rates and lifetimes of HONO and $NO_2$ from the Sheridan Fire. We observed the evolving plume structure of HONO and $NO_2$, as well as the $HONO/NO_2$ emission ratio that decreased in time and increased with sum FRP, similar to previous findings with $HONO/NO_2$ concentrations in Australian wildfires (Fredrickson et al., 2023). Using a 1-D horizontal Gaussian emission, first-order chemical model, we found that the EMG method to estimate emissions and lifetimes requires emissions from a fire to be relatively constant over several chemical and transport lifetimes. Idealized simulations of time varying fire emissions, transport and chemical loss illustrated the challenge of applying EMG to single overpass line density observations to derive emissions and lifetime estimates. Using Monte Carlo 1-D horizontal models developed with the diurnal FRP profile of the Sheridan Fire and the continuously sensed plume composition by the GCAS instrument, we provided constraints on the emission rate and lifetime of HONO and $NO_2$. The $NO_2$ lifetime is on the order of 20 to 40 minutes for this fire, which is on the lower end of reported wildfire $NO_2$ and $NO_x$ lifetimes. We hypothesize a large source of $RO_2$ drives the loss of $NO_x$. The HONO lifetime is longer than the expected clear-sky photolysis lifetime and consistent with a large fraction of the measured HONO column being in the core of the smoke plume with lower light levels and thus longer photolysis lifetimes. The HONO emissions using the Monte Carlo 1-D models were 5 to 10 times larger than that from both FINNv2.5 chemical mechanisms. On the other hand, the $NO_2$ emissions were larger than burned-area-based biomass burning emission inventories but were dwarfed by the FRP-based biomass burning emission inventories.

The Monte Carlo diurnal 1-D model procedure may not work well for all fires and has the following limitations. This procedure assumes that there is an individual fire with a constant-direction wind transporting the smoke plume. If there are multiple fires whose smoke plumes coalesce, this improved EMG will fail to estimate the true emission rate and lifetime, due to multiple point sources in different locations contributing their own unique time-varying emissions. These limitations apply to the regular EMG fit approach as well. In addition, trace-gas retrievals from UV-Vis measurements can have significant uncertainties arising from retrieval method and various auxiliary information such as a-priori vertical profile shapes, aerosols, etc. used in retrievals.

This study indicates the need for future research into HONO emissions quantification and for the impacts of HONO emissions on the chemistry downwind in a fire plume. We were only able to compare our HONO emissions to one biomass burning emission inventory, FINNv2.5, whereas we could compare $NO_2$ to at least four other inventories. Applying our methodology to hourly daytime observations of fires will improve the representation of HONO emissions from fires and therefore its impacts our understanding of fire plume reactive chemistry. With the launch of the TEMPO instrument in early April 2023, hourly daytime measurements of air pollutants are possible in the North American continent. Even though the

spatial resolution of TEMPO is coarser than GCAS, the loss of the fine structures in the sampled line densities is not critical to the techniques described in this paper. The Monte Carlo 1-D model output itself is smooth and lacks the noisy, fine features found in the GCAS line densities. This coarse representation may make finding the best model parameters easier with RMSE. Future research promises to narrow the uncertainty of wildfire emissions, emissions that evolve on a sub-hourly basis. The Monte Carlo diurnal 1-D model approach is primed for application on fires detected by TEMPO and the Geostationary Environment Monitoring Spectrometer.

**Appendix A: Plume rotation methodology**

To calculate the line densities of a sampled smoke plume, the smoke plume needs to be rotated such that the plume axis is parallel to the x-axis. Fig. A1 below summarizes our methodology of determining the plume axis and its rotation angle for all the Sheridan Fire smoke plumes. First, each data point of the regridded $NO_2$ VCDs from all GCAS-sampled tracks (tracks 10-14 and 17-20) are summed and then divided by the number of valid cells (non-NaN cells), producing a temporal average of the plume shape, since the averaging occurs over the different timestamps of the sampled tracks. Second, we calculate the background $NO_2$ VCD by averaging all cells upwind of the fire, Third, we create a plume mask, where only $NO_2$ VCDs that are greater than three times the background $NO_2$ VCD remain. These are the colored pixels in Fig. A1. Finally, we perform a linear regression on the plume $NO_2$ VCD, weighting the regression with the difference between the $NO_2$ VCDs and the maximum $NO_2$ VCD plume pixel. This weighting prioritizes the plume's edges, which define the plume shape. The angle of this linear fit becomes the plume rotation angle we apply to all the sampled plume tracks.

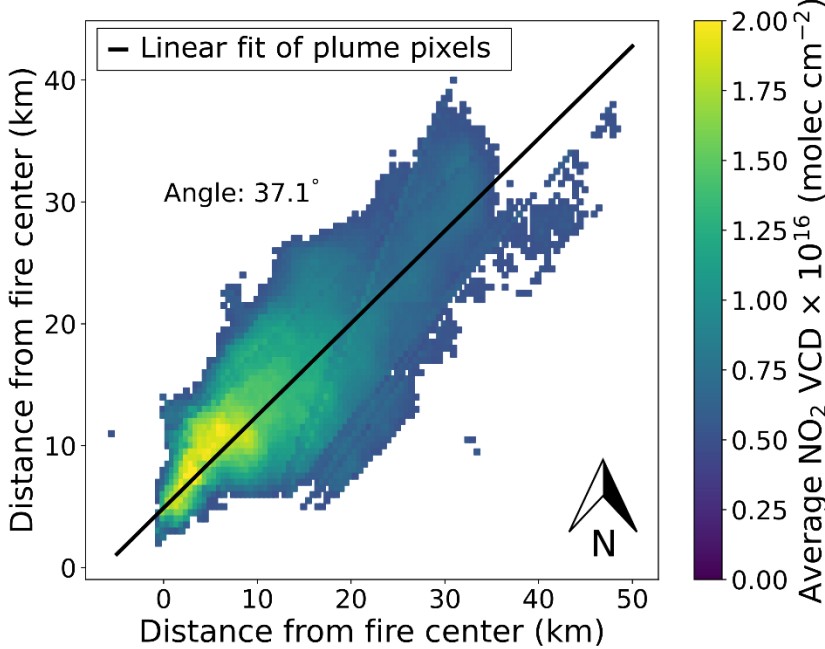

**Figure A1: NO₂ VCD plume averaged over GCAS sampling time, with a plume mask applied. The black line is a weighted linear fit of the plume pixels, where the plume edges have the higher weight. The angle of the line is displayed.**

**Appendix B: Idealized model simulation construction**

**Table B1: Detailed PECANS 1-D Model Configuration for Idealized Simulations**

| Model Parameter | Model Quantity |
| --- | --- |
| Model timestep | 1 s |
| Modeled run time | 10,000 s |
| Number of boxes in each dimension | x: 500 |
| | y: 0 |
| | z: 0 |
| Size of boxes in each dimension | x: 500 m |
| | y: 500 m |
| | z: 500 m |
| Transport scheme | Implicit2 |
| Wind type | Fixed |
| Wind speeds in each dimension | x: 9.88 m s⁻¹ |

|                                          |                                        |
|------------------------------------------|----------------------------------------|
|                                          | y: 0 m s$^{-1}$                        |
|                                          | z: 0 m s$^{-1}$                        |
| Diffusion coefficient in each dimension  | x: 100 m$^2$ s$^{-1}$                  |
|                                          | y: 0 m$^2$ s$^{-1}$                    |
|                                          | z: 0 m$^2$ s$^{-1}$                    |
| Chemical mechanism                       | Ideal first-order                      |
| Lifetime                                 | 1848 s                                 |
| Initial condition                        | Gaussian                               |
| Initial condition options                | center_x: 25,000 m                     |
|                                          | width_x: 3,000 m                       |
|                                          | height: 0 molec cm$^{-3}$             |
| Emission type                            | Gaussian                               |
| Emission options                         | center_x: 25,000 m                     |
|                                          | width_x: 6,770 m                       |
|                                          | total: 4.04e+25                        |
|                                          | molec s$^{-1}$                        |
| Model output frequency                   | 200 s                                  |

**Appendix C: Constraints on the Monte Carlo diurnal 1-D model method**

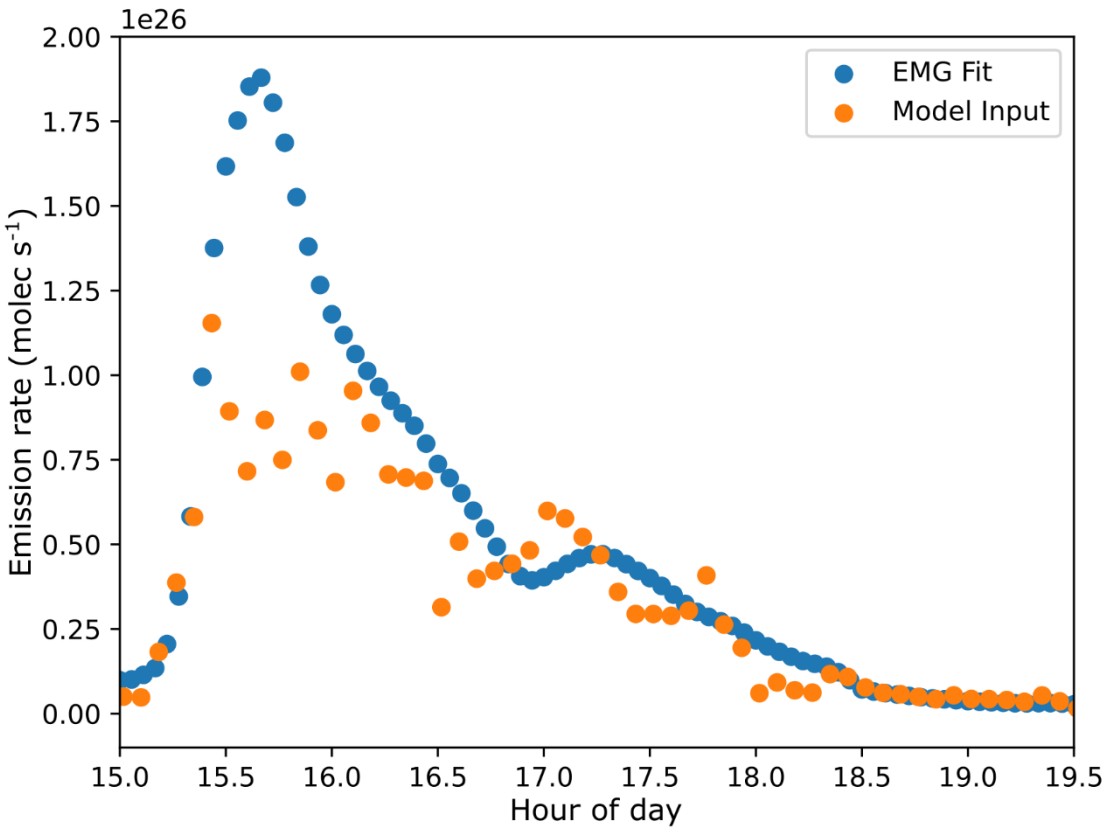

**Figure C1: Model input emission rate (molec s⁻¹) of a diurnal FRP profile source (orange) and the emission rate estimated from the EMG fit method applied to modeled line densities every 200 seconds (blue). The EMG fits overestimate the model input emission rate as the fire grows in output and lasts for approximately 40 minutes.**

**Appendix D: Best fit models from Monte Carlo diurnal 1-D model method**

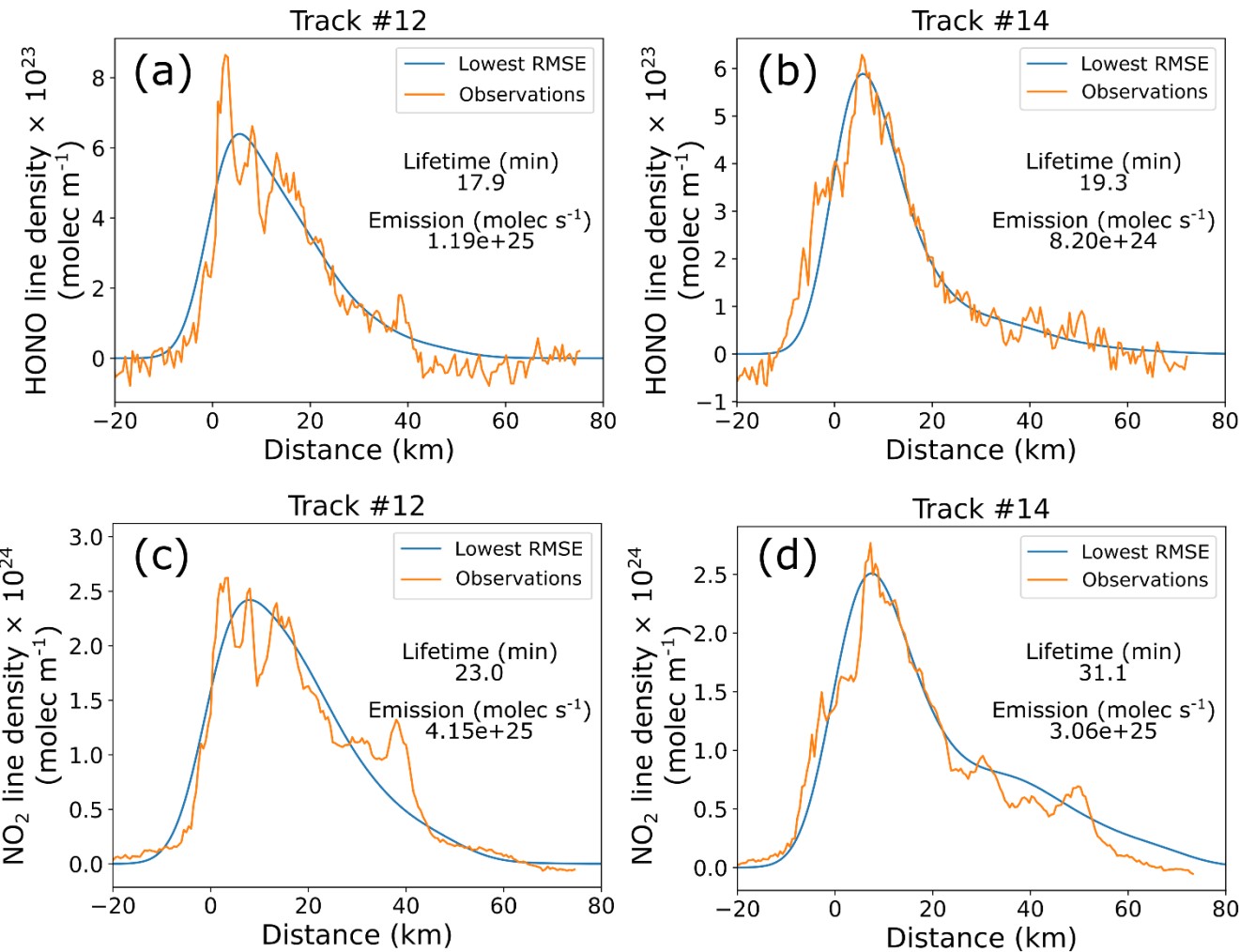

**Figure D1: Observed GCAS VCD line densities (orange) and best Monte Carlo diurnal 1-D model line densities (blue) for (a) HONO track 12, (b) HONO track 14, (c) NO₂ track 12, and (d) NO₂ track 14. Model lifetimes and emission rates are reported in the subfigures.**

**Appendix E: TUV Quick Calculator Inputs and Outputs**

**Table E1. TUV Quick Calculator Inputs**

| Inputs | Values | Justification |
| --- | --- | --- |
| Wavelength Start (nm) | 280 | Default |
| Wavelength End (nm) | 700 | Default |
| Wavelength Increments (-) | 420 | Default |

| | | |
|---|---|---|
| Latitude (°) | 34.68 | Sheridan Fire latitude |
| Longitude (°) | -112.89 | Sheridan Fire longitude |
| Date (YYYMMDD) | 20190816 | Analysis Date |
| Time (hh:mm:ss, GMT) | 22:45:15 and 23:20:30 | Track 12 and track 14 ER-2 overpass times |
| Overhead ozone column (du) | 300 | Default |
| Surface Albedo (0-1) | 0.15 | Default albedo of forests |
| Ground Elevation (km asl) | 1.55 | Sheridan Fire elevation |
| Measurement Altitude (km asl) | 6.25 | 4.7 km altitude |
| Clouds Optical Depth (-) | 0 | No clouds |
| Clouds Base (km asl) | N/A | No clouds |
| Clouds Top (km asl) | N/A | No clouds |
| Aerosols Optical Depth (-) | 0.5 | DC-8 DIAL measurement |
| Aerosols Single Scattering Albedo (SSA; -) | 0.7 | Default smoke SSA; 0.37 – 0.95 are valid values (Lewis et al., 2008; Liu et al., 2014) |
| Aerosols Alpha | 1.63 | (Saleh et al., 2013) |
| Sunlight Direct beam (-) | 1 | Default |
| Sunlight Diffuse down (-) | 1 | Default |
| Sunlight Diffuse up (-) | 1 | Default |

**Table E2. TUV Quick Calculator Outputs**

| Output Times | HONO photolysis frequency (s$^{-1}$) | HONO photolysis lifetime (min) |
|---|---|---|
| 22:45:15 | $1.467 \times 10^{-3}$ | 11.36 |
| 23:20:30 | $1.290 \times 10^{-3}$ | 12.92 |

**Code availability**

The PECANS model is available on GitHub (https://github.com/joshua-laughner/PECANS, last accessed 28 July 2024). This work used a customized version of PECANS v0.1.1; the customized version is available at https://doi.org/10.5281/zenodo.13621859.

**Data availability**

The FIREX-AQ aircraft datasets are available at https://www-air.larc.nasa.gov/cgi-bin/ArcView/firexaq and includes ER-2
GCAS data, the Fuel2Fire ecosystems-burned analysis, the Fuel2Fire GOES FRP diurnal cycle analysis, DC-8 DIAL-HSRL
data, DC-8 photolysis data, and DC-8 aircraft data (last access: 8 September 2024) (FIREX-AQ Science Team, 2019). The
ERA-5 reanalysis dataset is available at https://cds.climate.copernicus.eu/cdsapp#!/dataset/reanalysis-era5-pressure-levels?tab=overview (last access: 16 September 2024) (Hersbach et al., 2023). The GFED4s emission inventory data is
available at https://www.geo.vu.nl/~gwerf/GFED/GFED4/ (last access: 16 September 2024) (van der Werf et al., 2017). The
FINNv2.5 emission inventory data is available at https://rda.ucar.edu/datasets/d312009/ (last access: 16 September 2024)
(Wiedinmyer and Emmons, 2022). The QFEDv2.5 emission inventory data is available at
https://portal.nccs.nasa.gov/datashare/iesa/aerosol/emissions/QFED/v2.5r1/0.25/QFED/ (last access: 16 September 2024)
(Darmenov and da Silva, 2015). The GFASv1.2 emission inventory data is available at
https://ads.atmosphere.copernicus.eu/cdsapp#!/dataset/cams-global-fire-emissions-gfas?tab=overview (last access: 16
September 2024) (Kaiser et al., 2012).

**Author contribution**

CDF and JAT conceived of the project. CDF conducted the analysis and paper drafting. SJJ and LNL provided data and
input on results. UAJ processed and provided GFED4s fire emission data. JLL provided PECANS model code and input on
results. All authors contributed to manuscript review and editing.

**Competing interests**

At least one of the co-authors is a member of the editorial board of Atmospheric Measurement Techniques.

**Acknowledgements**

CDF acknowledges the National Aeronautics and Space Administration with the Future Investigators in NASA Earth and
Space Science and Technology (FINESST) Grant 80NSSC20K1612 for supporting this project. This research was supported
by NOAA Grant NA17OAR4310012 and NASA Grant 80NSSC23K092. A portion of this research was carried out at the Jet
Propulsion Laboratory (JPL), California Institute of Technology, under a contract with NASA (80NM0018D0004).
Government sponsorship is acknowledged. UAJ acknowledges NSF Division of Polar Programs (PLR) grants 1904128 and
2202287. This paper contains modified Copernicus Atmosphere Monitoring Service information 2018 and 2022. This paper
also contains modified Copernicus Climate Change Service information 2018 and 2022. Hersbach, H. et al. (2023) was

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
