# Peer review of "Remote Sensing Estimates of Time-Resolved HONO and NO2 Emission Rates and Lifetimes in Wildfires"

_Atmospheric Measurement Techniques, 2024_

## Referee Comment (RC1)

This manuscripts deals with airbone-based remote sensing data to characterize NO2 and HONO pumes from the Sheridan Fire. It uses auxiliary data also coming from airborne-based data, but also from satellite-based data. The exponentially modified gaussian (EMG) approach is used to extract the lifetimes and the emission rates from the NO2 and HONO emissions. Simulations regarding different plume cases are used to understand the limitations of the EMGs. A new methodology based on this very same approach is proposed, which results in a more consistent fit to the plume line densities. Comparison to the daily mean emissions of HONO and NO2 are also carried out. An important point of this manuscript is definitely the transfer of techniques developed in this work to geostationary space-based instruments such as TEMPO and GEMS, which can provide data with a very high temporal resolution.

I want to congratulate the authors for the enormous amount of work related to this manuscript. I also think the English is good, with almost no typos. Regarding the text, there are several comments on this document asking for more clarity. Some times it was difficult to keep up with the text, which made it difficult to read at some points. This also alludes to the science part, where there are some key concepts that were difficult for me to follow.

I consider this manuscript as an added-value to science that meets most quality standards. However, I believe that this work needs some corrections before publication, mostly for clarification. I have separated these corrections into 'major comments' and 'minor comments'.

**Major comments**

· Tests have been done using an airborne instrument. However, TEMPO and GEMS are space-based instruments. These instruments appear in the abstract and conclusions, and sometimes in the rest of the test. Regarding their importance for the future, it would be suitable to briefly explain how features as a different spatial resolution would affect the methodologies explain in this manuscript.

· Other methodologies for the estimation of emission rate the NO2 and HONO plumes can provide better results? Please, consider (for example) the IME method and the Cross Sectional Flux method. Maybe these methodology can suffer from incomplete plumes. However, due to the larger swath from space-based instruments, these instruments will be capable to capture entire plumes. As one of the most important points of this study is show the implementation of methodologies on airborne data to transfer them to satellite-based data in the future, it is important to deal with these questions.

· L375-383: A fit with higher R2 guarantees the most accurate values of lifetime and emission? I think it has more to do with the shape of the line density. If it has a gaussian shape it would be very suitable for the fit, but if not, then you can only try to make the best you can. What is then the meaning of the values with the maximum R2? Please, consider to develop more this concept.

**Minor comments**

**General comments**

· Please, check the acronym definitions and once the acronym is defined, used it. There are several cases in the text that are lately defined or where the acronym is not used.

· Please, make sure that the references to websites are correctly cited.

· Please, avoid unnecessary repetitions along the text.

**Abstract**:

· L22: Are these magnitudes really comparable? See comments later in the text.

1. **Introduction**:

· L35: Is this study only important for the US? If not, please consider to include a more global view in this sentence.

· L39: Regarding GFED and other inventories, the burned area is obtained from satellite-based measurements. Typically, a bottom-up approach is associated with ground-based, local, or detailed data collection at specific sites or sources (such as factories, vehicles, etc.). However, satellite data is more commonly associated with a top-down approach, where broad, large-scale observations (e.g., of emissions, land cover, or atmospheric concentrations) are made, and then models or algorithms are used to estimate emissions, trends, or impacts at finer scales. Please, consider checking the terminology of these approaches.

· L40: 'as a proxy for amount of material burned'… Please, try to avoid repetition for a more compact writing

· L55-56: Please, include references related to this statement.

· L78-79: Please, consider rewording this sentence for more clarity.

2. **Data and methods**

2.1. **FIREX-AQ**

· The section name refers to the FIREX-AQ campaign. However, the GCAS technical features, the AMF concept, the retrieval methodology and the a priori data are also explained. Please, consider to use different sections to describe the campaign and the methodology-related topics.

· L90: Is the second also measuring 'absolute nadir radiance'? Please, clarify.

· L93: A reference here for these instruments would be suitable.

· L93: Please, explain briefly in which aspects these instruments are similar.

· L98: The GCAS instrument measures radiance, from which the differential absorption spectra from a gas is deduced. Therefore, the different absorption spectra is not measured, but deduced. Then, I would consider removing the term 'measured'.

· L100: A figure showing an example of transmissivity spectra from NO2 and HONO would be illustrative to justify the selection of these fitting windows. Another option could be cite other works that used them.

·L105: AMF is not only dependent of the wavelength and altitude (first sentence), but also dependent of the solar zenith angle and the viewing zenith angle, among others. You refer to these parameters by mentioning the observation geometry (second sentence). However, consider rewording these sentences to build a more compact sentence where the dependencies of the AMF are not split in 2 sentences. It is not wrong, but I think this correction can provide more consistency in the AMF description.

· L108: Please, include a reference for VLIDORT.

· L132: Which is the range of the plume altitude? Is there such a big difference that can have an impact on the results or the selected value is well-suited for this purpose? For example, at different altitude values from the same plume, is there a significant difference in the preassure values? Please, justify the assumption taken regarding the altitude.

2.2. **ERA5**

· L138: Please, consider to show the pressure ranges presenting first the lower values and then the higher values.

2.3. **GOES FRP**

· L145: When reading this sentence, it seems that the main goal from these satellites are to locate fires. Please, consider rewording.

· L149: What is GeoMAC? Please, provide some clarification or at least a reference.

2.4. **Emission inventories**

· L159: In previous text, the angular degree unit was deg. Please, use just one of these units for consistency.

· L160: This sentences is something misleading. Then, this model can provide emission data for NO and HONO or only for NO? Regarding the way it is written, it seems that it says that HONO is a NOx, but it is not.

· L165: Which are the E units? Please, clarify.

· L170-173: Please, consider a simpler description.

· L185: Which are the data sources from GFASv1.2? All the data sources from the previous inventories were described.

2.5. **Analysis methods**

2.5.1. **Calculation of HONO and NO2 line densities**

· L196: This can work for gaussian plume shapes as the ones you show in this study. However, if the direction of the plume is extracted from the plume shape, it will not always align with the wind direction. Please, consider this for the discussion.

· L196: Time averaged VCD plume? Please, clarify.

· Figure 1: 2 of these panels are already shown in Figure 2. You could consider remove them from here or from Figure 1. If you keep the panels from this figure, you can allude to Figure 1 to refer to the track 14 results.

· Figure 1: Please, consider a more compact caption.

· L206: trackS

 · Appendix A: In my opinion, it is very difficult to understand this. Please, consider rewording to solve this.

· Appendix A: Please, consider to add some text in this appendix for clarification.

2.5.2. **Exponentially Modified Gaussian (EMG)**

· L217: NOx = NO + NO2. Then, NO2 repeated?

3. **Results and discussion**

3.1. **HONO and NO2 plume structure in the Sheridan Fire**

· Why there is a better resolution for NO2 maps? Please, clarify.

· L264-265: '... in all three overpasses, HONO and NO2 share local maxima, plume edges, and plume shape'. Please, consider a discussion about this result. What does this mean?

**3.2. EMG emission rates and lifetimes from the Sheridan Fire**

· L274: I would avoid to use the term 'step function' when you are referring to the extracted line densities as they are not extracted according to a known function. Please, consider to use other terms as 'drop sharply' or similar.

· L291-293: This sentence is somewhat confusing in expressing the timeline of events. Please, consider to reword this.

· L294: You could avoid to refer to Fig. 3f if you are already alluded to 'track 14'.

· L301: If you show the ratio of emission rates, you should first calculate them individually. Please, consider to include this calculation in the text (or if not, in an appendix).

· L301: What do you mean with 'similar sampling biases'? Please, clarify.

· L303: '... less HONO is being emitted than NO2.…' Is this correct? As a I understand, it makes more sense to me 'a relative decrease of HONO in reference to previous values' instead of 'less HONO is being emitted than NO2'. 'less HONO is being emitted than NO2' is always true as the ratio is lower than 1. Please, clarify.

· L306: As I understand, constant emission factor ratios should lead to constant emission ratios. To make this clear, I would change here 'emission factor ratio' to 'emission ratio'.

· Figure 4: How you can extract the emission rates of HONO and NO2 from 7 points (Figure 4) regarding that you only have 3 plumes (3 tracks) (L204-207)

**3.3. An improved EMG methodology: Monte Carlo diurnal 1-D models**

· L353-355: Please, add a reference related to this idea. Why an increase of thermal ouput lead to an underestimation of the emission rate? Please, clarify.

· L378: From which time? Please, clarify.

· L375-383: As I understand, here you are refering to the 4th PECAN 1-D configuration. Please, clarify this in the text.

· L378: See above where? Please, clarify.

· L379: Where are these ranges coming from exactly?

· L3979: As written here, it seems that the lifetime is also included in the division. Please, consider to reword this sentence to better readability.

· L375-383: Please, specify at which data this exercise is going to be applied.

· L381: PECANS

· L375-L383: This paragraph needs further improvement as it deals with complex concepts that need some clarification.

· L385: Here, the plotted value is R2. You could say that, regarding a R2 value, you could find that it is approximately constant in a combination of lifetime and emission rate values that follow an inverse relationship. However, you cannot say that 'the emission rate has an inverse relationship with lifetime'. Please, reword this sentence.

· Figure 6: Why are two y-axis showing the emission rate in two different units? Please, consider just using one and write to the relation between these 2 units in the text (if needed).

· Figure 6: There are 2 colorbars that are representing the same value. Please, consider to keep only one.

· L400: Which are the input values from the TUV Quick Calculator? Please, clarify.

· L424-426: What does this mean? A presence of a large source of RO2 is realistic? If it is not, what you can say about it? Looking at the plume from Figure A1, it can be seen that there is some influence from the surface. Can the surface have an impact on the retrieval and therefore in the lifetime? Please, clarify.

**3.4. **Biomass burning emission inventories underestimate observationally constrained emission estimates**

· L434: Please, consider previous comments regarding the bottom-up and top-down approaches.

· Figure 7: In this section, you deal with the minimum and maximum values of emission rate extracted from track #12 and #14 as representative of the whole day. You called them as 'daily emission rate', which is not true because it does not consider the emission rate at different times during the day. Please, consider to change the terminology and to develop a discussion with this change. Stating that there are biases in reference to daily mean emission from inventories can be misleading.

---

## Author Comment (AC1)

Dear Editor,

*Referee #2's comments are in italic* and our answers to all comments of reviewer #1 are embedded in red. **Bold text** are manuscript additions and  are manuscript deletions.

*The authors evaluate the emission rate and lifetime of NO2 and HONO during the FIREX-AQ campaign. The authors use aircraft and GOES-16 and -17 observations to do the analysis. The following comments are provided to clarify the context for the reader's better understanding. Among the comments, one of the significant issues is the suitability of using GOES-17 observations in the study due to mechanical issues of ABI. Without evidence or references that GOES-17 FRP products are safely usable, erroneous signals from malfunctioning GOES-17 ABI may affect the study's results.*

The authors thank Referee #2 for reading and evaluating our manuscript. Our responses to your comments are below.

**Major comments**

1. *In lines 104 – 115, the authors did not describe how to calculate the AMF. If it is the author's intent not to describe in detail, references are needed to give readers a proper way to calculate the values. In addition, in line 110, please indicate the bidirectional reflectance distribution functions are used to describe the surface. The optical properties of aerosols also weren't mentioned in the context.*

   To address the deficiencies in the descriptions of the AMF calculations, BRDFs, and aerosol optical properties, we have made the following additions to lines 102 – 111:

   "**As discussed in Lamsal et al. (2017), since the retrievals use average radiance from a clean background (reference location) due to the lack of solar irradiance measurements for normalization, the spectral fitting procedure provides differential slant column amounts which represent slant columns with respect to the reference location.**

   **V**ertical column densit**ies**  **below aircraft are calculated**  **using** the **differential** SCD**s**  and air mass factor**s** (AMF**s**) **following the approach discussed in Lamsal et al. (2017)**. …**Impact of aerosols is partially accounted for in the retrievals due to the use of average radiance measurements for the spectral fit as well as reference correction in the calculation of VCDs; retrievals are likely affected for high aerosol cases.**"

2. *Section 2.3 describes FRP derived from GOES-16 and -17 observations. However, as mentioned in https://www.goes-r.gov/users/GOES-17-ABI-Performance.html, GOES-17 ABI has cooling system issues. Please provide evidence or references indicating GOES-17 ABI observations can be used to obtain FRP. In addition, in line 149, why choose 5 %?*

   While a research review could not find an article about the use of GOES-17 ABI during the time period of the FIREX-AQ campaign, J. V. Hall et al. published a paper on June 6, 2023 about validating geostationary active fire products from GOES-17 ABI, GOES-16 ABI, and Himawari AHI. They conducted their assessments during the winter and summer seasons and found

comparable active fire pixel detection rates between the NOAA (GOES) ABI Fire Detection and Characterization product and the EUMETSAT FRP-PIXEL product. While this paper implemented a different algorithm than used during FIREX-AQ, it is based on the WFABBA algorithm used to derive the FRPs for FIREX-AQ.

Many papers covering the FIREX-AQ campaign use this GOES-17 FRP data in their research (Peterson et al., 2022; Warneke et al., 2023; Wiggins et al., 2020; Wiggins et al., 2021). Additionally, we use the same dataset in both Wiggins papers. To be consistent with the previous research concerning this field campaign, we are still opting to use the GOES-17 FRP data processed by the FIREX-AQ team. However, we have added a statement in the manuscript to acknowledge the GOES-17 ABI cooling system issues, presented below:

"FRP information was retrieved from both GOES-16 and GOES-17 using the WildFire Automated Biomass Burning Algorithm from the University of Wisconsin, Madison (Schmidt, 2020). **We acknowledge that the GOES-17 ABI has cooling system issues and thus impacts the FRP retrievals during this time period. We have opted to keep the GOES-17-derived FRPs to be consistent with previous FIREX-AQ analyses that use GOES-17 data (Peterson et al., 2022; Warneke et al., 2023; Wiggins et al., 2020, 2021)."**

We used 5% because we used the data hosted on the FIREX-AQ data repository (FIREX-AQ, 2019). This methodology detailing 5% is described in the FIREXAQ-Fuel2Fire-GOESDiurnaCycle_Analysis_R1_ReadMe.docx file found at this link: https://www-air.larc.nasa.gov/cgi-bin/ArcView/firexaq?ANALYSIS=1.

The following references were added to the References section:

[revised manuscript text omitted]

3. *The Gaussian emission rate is shown in Fig. B1 and Figure 5 shows the results. These two figures can be put together, and the constant and step-change emission rates can be added to Figure 5 for better understanding.*

We have amended Fig. 5 to include the emission profiles and removed Fig. B5. The new figure is shown below:

[Figure]

"Figure 5: **(a) The emission profile used as input to the idealized 1-D constant emissions model.** (b~a~) Line densities of a hypothetical compound from ~a~the 1-D model with prescribed chemical lifetime of 30.8 minutes **(black, dashed line)** and constant emissions colored by model**ed** run time. (c~b~) EMG fit / model prescribed emission rate (blue) and lifetime (light blue) at every model output step through the course of the entire model**ed** run time. The model**ed** time equal to the model lifetime is indicated with a vertical, black dashed line. **(d) The emission profile used**

**as input to the idealized 1-D step-change emissions model.** (e~c~) Line densities ~for a~ **from the** 1-D model where the emission rate is halved halfway through the model**ed** run time, colored by model**ed** run time. (f~d~) Similar to (c~b~), but for the model in (e~c~). The vertical, gold dashed line indicates where the emissions were halved. **(g) The emission profile used as input to the idealized 1-D Gaussian emissions model.** (h~e~) Line densities ~for a~ **from the** 1-D model where the emissions are scaled to a Gaussian PDF with a mean of 5000 and standard deviation of 1000, colored by model**ed** run time. (i~f~) Similar to (c~b~), but for the model in (h~e~). The vertical, gold dashed line indicates peak Gaussian emissions."

4. *Figure 6 and lines 375 – 393 describe the results based on $R^2$ Since the authors compared the model and satellite observations, would it be better to compare results using biases, standard deviations, or root-mean-square errors?*
   We agree with the reviewer, as well as other reviewers that described this same issue. We now describe the results based on root mean square errors. Please refer to the response to RC1 major comments for the full list of changes to the figures and text.

**Minor comments**

1. *In lines 35 – 36, references are needed to the statement, "As the intensity of fires and burned area from fires are predicted to increase in the United States in the future."*

   We have rephrased this statement and added references to support this statement:

   "As the intensity of fires and burned area from fires ~are predicted to increase~ in the United States **and globally have had increasing trends over the past few decades and are predicted to increase** in the future **(Barbero et al., 2015; Burton et al., 2024; Cunningham et al., 2024; Dennison et al., 2014)**,…"

   The following references were added to the References section:

   "**Barbero, R., Abatzoglou, J. T., Larkin, N. K., Kolden, C. A., and Stocks, B.: Climate change presents increased potential for very large fires in the contiguous United States, Int. J. Wildland Fire, 24, 892, https://doi.org/10.1071/WF15083, 2015.**

   **Burton, C., Lampe, S., Kelley, D. I., Thiery, W., Hantson, S., Christidis, N., Gudmundsson, L., Forrest, M., Burke, E., Chang, J., Huang, H., Ito, A., Kou-Giesbrecht, S., Lasslop, G., Li, W., Nieradzik, L., Li, F., Chen, Y., Randerson, J., Reyer, C. P. O., and Mengel, M.: Global burned area increasingly explained by climate change, Nat. Clim. Chang., 14, 1186–1192, https://doi.org/10.1038/s41558-024-02140-w, 2024.**

   **Cunningham, C. X., Williamson, G. J., and Bowman, D. M. J. S.: Increasing frequency and intensity of the most extreme wildfires on Earth, Nat Ecol Evol, 8, 1420–1425, https://doi.org/10.1038/s41559-024-02452-2, 2024.**

   **Dennison, P. E., Brewer, S. C., Arnold, J. D., and Moritz, M. A.: Large wildfire trends in the western United States, 1984-2011, Geophys. Res. Lett., 41, 2928–2933, https://doi.org/10.1002/2014GL059576, 2014.**"

2. *In lines 42 – 44 and 154 - 155, the way of citing references is confusing. Please put matched references after the name of the relevant inventories.*

The cited references are now listed after the inventory is named, as demonstrated below:

"Some examples of burned area inventories are the Global Fire Emissions Database (GFED**; Giglio et al., 2013)** and the Fire INventory from NCAR (FINN) (Wiedinmyer et al., 2011). Some examples of FRP-derived inventories are the Quick Fire Emissions Dataset (QFED**; Darmenov and da Silva, 2015**) and the Global Fire Assimilation System (GFAS) ( Kaiser et al., 2012)."

"The derived emission rates from this work are compared to a set of commonly used biomass burning emission inventories, including GFED4s **(Giglio et al., 2013; van der Werf et al., 2017)**, FINNv2.5 **(Wiedinmyer et al., 2023)**, QFEDv2.5 **(Darmenov and da Silva, 2015)**, and GFASv1.2 ( Kaiser et al., 2012)."

3. *In lines 76 – 78 and after, please provide references to GCAS and FIREX-AQ, and also provide references to TROPOMI and TEMPO instruments in line 93.*

The cited references are now included in the main text, as demonstrated below:

"These measurements were made using the GeoCAPE Airborne Simulator (GCAS**; Kowalewski and Janz, 2014**) instrument aboard the National Aeronautics and Space Administration (NASA) ER-2 aircraft for the Fire Influence on Regional to Global Environments and Air Quality (FIREX-AQ**; Warneke et al., 2023**) campaign."

"The GCAS instrument shares similar design specifications with the **TROPOspheric Monitoring Instrument (**TROPOMI**; Veefkind et al., 2012)** and **Tropospheric Emissions: Monitoring of Pollution (**TEMPO**; Chance et al., 2013)** instruments."

The references below have been added to the References section:

"**Chance, K., Liu, X., Suleiman, R. M., Flittner, D. E., Al-Saadi, J., and Janz, S. J.: Tropospheric emissions: monitoring of pollution (TEMPO), SPIE Optical Engineering + Applications, San Diego, California, United States, 88660D, https://doi.org/10.1117/12.2024479, 2013.**

**Veefkind, J. P., Aben, I., McMullan, K., Förster, H., De Vries, J., Otter, G., Claas, J., Eskes, H. J., De Haan, J. F., Kleipool, Q., Van Weele, M., Hasekamp, O., Hoogeveen, R., Landgraf, J., Snel, R., Tol, P., Ingmann, P., Voors, R., Kruizinga, B., Vink, R., Visser, H., and Levelt, P. F.: TROPOMI on the ESA Sentinel-5 Precursor: A GMES mission for global observations of the atmospheric composition for climate, air quality and ozone layer applications, Remote Sensing of Environment, 120, 70–83, https://doi.org/10.1016/j.rse.2011.09.027, 2012.**

**Warneke, C., Schwarz, J. P., Dibb, J., Kalashnikova, O., Frost, G., Al-Saad, J., Brown, S. S., Brewer, Wm. A., Soja, A., Seidel, F. C., Washenfelder, R. A., Wiggins, E. B., Moore, R. H., Anderson, B. E., Jordan, C., Yacovitch, T. I., Herndon, S. C., Liu, S., Kuwayama, T., Jaffe, D., Johnston, N., Selimovic, V., Yokelson, R., Giles, D. M., Holben, B. N., Goloub, P., Popovici, I., Trainer, M., Kumar, A., Pierce, R. B., Fahey, D., Roberts, J., Gargulinski, E. M., Peterson, D. A.,**

Ye, X., Thapa, L. H., Saide, P. E., Fite, C. H., Holmes, C. D., Wang, S., Coggon, M. M., Decker, Z. C. J., Stockwell, C. E., Xu, L., Gkatzelis, G., Aikin, K., Lefer, B., Kaspari, J., Griffin, D., Zeng, L., Weber, R., Hastings, M., Chai, J., Wolfe, G. M., Hanisco, T. F., Liao, J., Campuzano Jost, P., Guo, H., Jimenez, J. L., Crawford, J., and The FIREX-AQ Science Team: Fire Influence on Regional to Global Environments and Air Quality (FIREX-AQ), JGR Atmospheres, 128, e2022JD037758, https://doi.org/10.1029/2022JD037758, 2023.”

4. *Please provide the full name of TROPOMI in line 93, as it is its first time mentioned in the main context.*

We thank you for catching this error. We also realized that this sentence is also the first time TEMPO is mentioned in the main text. This sentence has been amended to the following:

“The GCAS instrument shares similar design specifications with the **TROPOspheric Monitoring Instrument (**TROPOMI**) and Tropospheric Emissions: Monitoring of Pollution (**TEMPO**)** instruments.”

We also removed the full names of TROPOMI and TEMPO later in the main text:

“One such approach relies on fitting an EMG to the line density of, for example, daily satellite observations of $NO_2$ columns from TROPOMI (Jin et al., 2021).”

“With the launch of **the** TEMPO instrument in early April 2023, hourly daytime measurements of air pollutants are possible in the North American continent.”

5. *Please provide references to VLIDORT in line 108.*

We have added a reference to VLIDORT in line 108, amended as follows:

“In this study, AMFs are calculated using the vector linearized discrete ordinate radiative transfer code (VLIDORT)**, version 7.2 (Spurr, 2006)**.”

The reference below has been added to the References section:

“**Spurr, R. J. D.: VLIDORT: A linearized pseudo-spherical vector discrete ordinate radiative transfer code for forward model and retrieval studies in multilayer multiple scattering media, Journal of Quantitative Spectroscopy and Radiative Transfer, 102, 316–342, https://doi.org/10.1016/j.jqsrt.2006.05.005, 2006.**”

6. *Please provide references to Carnegie-Ames-Stanford Approach model in line 157.*

We have added a reference to the Carnegie-Ames-Stanford Approach model in line 157, amended as follows:

“…with the Carnegie-Ames-Stanford Approach model, which estimates fuel loads and combustion completeness **(Potter et al., 1993)**.”

The reference below has been added to the References section:

"**Potter, C. S., Randerson, J. T., Field, C. B., Matson, P. A., Vitousek, P. M., Mooney, H. A., and Klooster, S. A.: Terrestrial ecosystem production: A process model based on global satellite and surface data, Global Biogeochemical Cycles, 7, 811–841, https://doi.org/10.1029/93GB02725, 1993.**"

7. *Please check the units "-m" in lines 156 and 161.*

The units in lines 156 and 161 now read as "500 m" and "375 m" respectively, removing the dashes.

8. *In lines 166 – 167, 180 – 181, and 229 - 232, please provide units to the variables.*

While a combination of units is possible for these variables, we have provided the reader guideline units if they wanted to reproduce our methodology as follows:

"where $i$ is a specific compound, $E$ is the emissions **(g)**, $A$ is the area burned **(m$^2$)**, $B$ is the amount of biomass **(kg m$^{-2}$)**, $FB$ is the fraction of biomass burned **(unitless)**, and $EF$ is the emission factor with units of mass of $i$ per mass biomass burned **(g kg$^{-1}$)**."

"…where $E_i$ is the emission rate of compound *i* per unit area **(g m$^{-2}$ s$^{-1}$)**, $\alpha$ is a constant that relates time integrated FRP (fire radiative energy) to dry biomass burned **(kg J$^{-1}$)**, $EF_i$ is the emission factor of compound *i* **(g kg$^{-1}$)**, and $A$ is the area of the satellite pixel **(m$^2$)**."

"From the EMG parameters, an emission rate ($E_{EMG}$; **molec s$^{-1}$**) and effective lifetime ($\tau_{EMG}$; **s**)…"

9. *Please provide references to SAPRC99, MOZART, and GOES-Chem in lines 171 - 172.*

We have added references to SAPRC99, MOZART, and GOES-Chem in the main text as follows:

"Statewide Air Pollution Research Center Mechanism (SAPRC99**; Carter, 1999**), Model for Ozone and Related chemical Tracers (MOZART**; Emmons et al., 2020**), and Goddard Earth Observing System with Chemistry (GOES-Chem**; Bey et al., 2001**)."

The references below have been added to the References section:

"**Bey, I., Jacob, D. J., Yantosca, R. M., Logan, J. A., Field, B. D., Fiore, A. M., Li, Q., Liu, H. Y., Mickley, L. J., and Schultz, M. G.: Global modeling of tropospheric chemistry with assimilated meteorology: Model description and evaluation, J. Geophys. Res., 106, 23073–23095, https://doi.org/10.1029/2001JD000807, 2001.**

**Carter, W. P. L.: Documentation of the SAPRC-99 Chemical Mechanism for VOC Reactivity Assessment, University of California, Riverside, 1999.**

**Emmons, L. K., Schwantes, R. H., Orlando, J. J., Tyndall, G., Kinnison, D., Lamarque, J., Marsh, D., Mills, M. J., Tilmes, S., Bardeen, C., Buchholz, R. R., Conley, A., Gettelman, A., Garcia, R., Simpson, I., Blake, D. R., Meinardi, S., and Pétron, G.: The Chemistry Mechanism in the Community Earth System Model Version 2 (CESM2), J Adv Model Earth Syst, 12, e2019MS001882, https://doi.org/10.1029/2019MS001882, 2020.**"

10. *In line 214, please define OMI.*

We thank you for catching this error. This sentence has been amended to the following:

"EMGs have been applied to satellite observations of **the Ozone Monitoring Instrument (**OMI**)** and TROPOMI NO$_2$ to estimate emissions and lifetimes from point sources and urban areas (Beirle et al., 2011; De Foy et al., 2015; Goldberg et al., 2019; Laughner and Cohen, 2019; Lu et al., 2015; Xue et al., 2022)."

11. *Please add the direction of North as Fig 1 to Fig 2 and indicate the flight track number for the corresponding Fig 2 Figures.*

We have amended Fig. 2 with requests from the reviewer. The updated Fig. 2 within the manuscript is shown below:

[Figure]

12. *In line 264, please describe what enhanced NO2 is.*

We have amended the main text to define enhanced NO$_2$ earlier in the main text, in Section 2.5.1. Calculation of HONO and NO$_2$ line densities as follows:

"We then subtracted from the entire scene the average HONO and NO$_2$ VCDs upwind of the fire**, called the background HONO and NO$_2$ VCDs, resulting in enhanced HONO and NO$_2$ VCDs solely from the wildfire. This process also removes any stratospheric component of HONO and NO$_2$.**"

---

## Author Comment (AC2)

Dear Editor,

*Referee #1's comments are in italic* and our answers to all comments of reviewer #1 are embedded in red. **Bold text** are manuscript additions and  are manuscript deletions.

*This manuscripts deals with airbone-based remote sensing data to characterize NO2 and HONO pumes from the Sheridan Fire. It uses auxiliary data also coming from airborne-based data, but also from satellite-based data. The exponentially modified gaussian (EMG) approach is used to extract the lifetimes and the emission rates from the NO2 and HONO emissions. Simulations regarding different plume cases are used to understand the limitations of the EMGs. A new methodology based on this very same approach is proposed, which results in a more consistent fit to the plume line densities. Comparison to the daily mean emissions of HONO and NO2 are also carried out. An important point of this manuscript is definitely the transfer of techniques developed in this work to geostationary space-based instruments such as TEMPO and GEMS, which can provide data with a very high temporal resolution.*

*I want to congratulate the authors for the enormous amount of work related to this manuscript. I also think the English is good, with almost no typos. Regarding the text, there are several comments on this document asking for more clarity. Some times it was difficult to keep up with the text, which made it difficult to read at some points. This also alludes to the science part, where there are some key concepts that were difficult for me to follow.*

*I consider this manuscript as an added-value to science that meets most quality standards. However, I believe that this work needs some corrections before publication, mostly for clarification. I have separated these corrections into 'major comments' and 'minor comments'.*

***Major comments***

- *Tests have been done using an airborne instrument. However, TEMPO and GEMS are space-based instruments. These instruments appear in the abstract and conclusions, and sometimes in the rest of the test. Regarding their importance for the future, it would be suitable to briefly explain how features as a different spatial resolution would affect the methodologies explain in this manuscript.*

  There is not a substantial difference in the methodology if using space-based or airborne instruments. The methodology requires the retrieval of VCDs. Each instrument will have a different sampling resolution and thus different grid resolution of these final products. That is the only difference concerning the application of this methodology. They employ different procedures to retrieve VCDs, but an instrument comparison of VCD retrievals is beyond the scope of this paper.

  We have added the following text to the "Analysis methods" section and the "Conclusions" section of the manuscript, separately, expanding upon how the methodologies would differ for the space-based instruments:

  **"…For use with an hourly space-based instrument like TEMPO, only the regridding resolution would differ in this procedure (0.02° for TEMPO). This lower resolution in grid size results in a lower resolution in line density, where the fine structures seen with the GCAS instrument will not be as resolved…"**

**"…**With the launch of **the** TEMPO instrument in early April 2023, hourly daytime measurements of air pollutants are possible in the North American continent. **Even though the spatial resolution of TEMPO is coarser than GCAS, the loss of the fine structures in the sampled line densities is not critical to the techniques described in this paper. The Monte Carlo 1-D model output itself is smooth and lacks the noisy, fine features found in the GCAS line densities. This coarse representation may make finding the best model parameters easier with RMSE…"**

- *Other methodologies for the estimation of emission rate the NO2 and HONO plumes can provide better results? Please, consider (for example) the IME method and the Cross Sectional Flux method. Maybe these methodology can suffer from incomplete plumes. However, due to the larger swath from space-based instruments, these instruments will be capable to capture entire plumes. As one of the most important points of this study is show the implementation of methodologies on airborne data to transfer them to satellite-based data in the future, it is important to deal with these questions.*
  We have included a brief discussion of a few other emission estimations methods, such as the IME and Cross-Sectional Flux methods. As this paper only evaluates the EMG method and improves upon it, it is beyond the scope of this paper to evaluate two more methodologies. The primary point of this paper was to respond to two papers using TROPOMI and the EMG method to make emissions estimates. The discussion in the manuscript is presented below:

  **"The EMG is fit to a plume and the enhancement factor from the fitting procedure is directly linked to the emissions. Other methods are also used to estimate emissions from satellites, such as the integrated mass enhancement (IME) method, which multiplies the wind speed by the integrated vertical column densities (VCDs), the cross-sectional flux method (CFM), which estimates emissions by averaging the flux through multiple cross sections perpendicular to the plume direction, and Chemical Transport Models (CTMs), which can predict emissions more accurately but are computationally expensive. While these other techniques are proven to be useful, this paper focuses on dissecting the EMG approach.** Forming conclusions using daily observations of fires, while a good starting point, does not capture the diurnal variability that fires exhibit (Wiggins et al., 2020)."

- *L375-383: A fit with higher R2 guarantees the most accurate values of lifetime and emission? I think it has more to do with the shape of the line density. If it has a gaussian shape it would be very suitable for the fit, but if not, then you can only try to make the best you can. What is then the meaning of the values with the maximum R2? Please, consider to develop more this concept.*
  Upon further review of the use of $R^2$ and other reviewer feedback, we have changed the evaluation metric to be root mean square error (RMSE). This evaluation metric is optimal for Gaussian errors (Hodson, 2002).

  The new Figure 6, Figure 7, and Figure D1 are below. Note: the best model runs did not change when using RMSE as the evaluation metric:

[Figure]

"Figure 6: The R**MSE2** value between 100 PECANS models with varied total emission rate and lifetime model input parameters and the observed GCAS HONO VCD line density for (a) track 12 and (b) track 14. **The emission rate is plotted by multiplying the total emission rate by the normalized FRP at the time of each track.** A filled contour map creates a 2D interpolation of the randomly sampled models (**dots**). (c) and (d) Similar to (a) and (b), but for GCAS NO2 VCD."

[Figure]

"Figure 7: Daily mean emissions of (a) HONO and (b) NO₂ from the Monte Carlo diurnal 1-D model method (MC diurnal 1-D) using the **lowest RMSE model for each track** to define the  emissions, and the following biomass burning emission inventories: FINNv2.5 SAPRC, FINNv2.5 MOZART, GFED4s, FINNv2.5, QFEDv2.5, and GFASv1.2."

[Figure]

"Figure D1: Observed GCAS VCD line densities (orange) and best Monte Carlo diurnal 1-D model line densities (blue) for (a) HONO track 12, (b) HONO track 14, (c) $NO_2$ track 12, and (d) $NO_2$ track 14. Model lifetimes and emission rates are reported in the subfigures."

The following text in the manuscript has been updated:
"…The  **root mean square error** **(RMSE)** between the model and the  **GCAS** observations for **tracks 12 and 14**  **are** shown in Fig. 6 for HONO and $NO_2$. The absolute  **minimum RMSE** provides an indication of the most likely lifetime and emissions rate combination(s)…"

"…In Fig. 6, **RMSE is approximately constant in a combination of lifetime and emission rate values that follow an inverse relationship**, which is expected since general mass balance constitutes that concentration is proportional to emissions and inversely proportional to loss rate. In Figs. 6a and 6b, the **smallest RMSE** between the HONO observed and model derived line densities occur with a lifetime between 15 and 27 minutes and an emissions rate of 0.6 to 1.**75** × $10^{25}$ molec s⁻¹. In Figs. 6c and 6d, $NO_2$ has a larger viable range of lifetimes and total emissions rates, which range between  **20** and 45 minutes and 2.0 to 5.**5** × $10^{25}$ molec s⁻¹, respectively. Track 18 was excluded from this analysis as measurements occurred after the Sheridan Fire had ceased its high emissions for the day. The model runs with the **smallest RMSE** values are shown in Fig. D1…"

"…As shown in Fig. 7a, FINNv2.5 HONO emissions for the SAPRC and MOZART mechanisms underestimate the **track 12 and track 14** daily mean emission rate of our method (MC diurnal 1-D) by a factor of 5 **9.4** for  **track 12** and a factor of  **7.2** for **track 14**…"

Response Reference:
Hodson, T. O.: Root-mean-square error (RMSE) or mean absolute error (MAE): when to use them or not, Geosci. Model Dev., 15, 5481–5487, https://doi.org/10.5194/gmd-15-5481-2022, 2022.

*Minor comments*

1. **General comments**
   a. *Please, check the acronym definitions and once the acronym is defined, used it. There are several cases in the text that are lately defined or where the acronym is not used.*
      We have corrected our acronym usage in some instances. However, we believe it is useful to define an acronym that is commonly used in an area of research. This makes searching through text easier for those readers trying to find key words.

   b. *Please, make sure that the references to websites are correctly cited.*
      All websites in the main text of the manuscript have been replaced with citations to the software user manuals.

   c. *Please, avoid unnecessary repetitions along the text.*
      We have read through the text and have removed unnecessary repetitions. However, some repetitions have remained to benefit the reader.

2. **Abstract**
   a. *L22: Are these magnitudes really comparable? See comments later in the text.*
      These quantities are comparable. We have clarified in the abstract what we are comparing by adding the following text:

      "We assess the validity of a range of emission rate and lifetime combinations for both HONO and NO2 as the fire evolves by comparing the resulting line density predictions to the **line density** observations."

3. **Introduction**
   a. *L35: Is this study only important for the US? If not, please consider to include a more global view in this sentence.*
      This study is not only important for the US. To express that this study has global implications, we have amended the manuscript with the following text:

      "As the intensity of fires and burned area from fires  in the United States **and globally have had increasing trends over the past few decades and are predicted to increase** in the future **(Barbero et al., 2015; Burton et al., 2024; Cunningham et al., 2024; Dennison et al., 2014)**, it is important to understand how wildfire emissions change with fire properties and how these emissions impact local, regional, and global atmospheric composition."

      The following references were added to the References section:

"Barbero, R., Abatzoglou, J. T., Larkin, N. K., Kolden, C. A., and Stocks, B.: Climate change presents increased potential for very large fires in the contiguous United States, Int. J. Wildland Fire, 24, 892, https://doi.org/10.1071/WF15083, 2015.

Burton, C., Lampe, S., Kelley, D. I., Thiery, W., Hantson, S., Christidis, N., Gudmundsson, L., Forrest, M., Burke, E., Chang, J., Huang, H., Ito, A., Kou-Giesbrecht, S., Lasslop, G., Li, W., Nieradzik, L., Li, F., Chen, Y., Randerson, J., Reyer, C. P. O., and Mengel, M.: Global burned area increasingly explained by climate change, Nat. Clim. Chang., 14, 1186–1192, https://doi.org/10.1038/s41558-024-02140-w, 2024.

Cunningham, C. X., Williamson, G. J., and Bowman, D. M. J. S.: Increasing frequency and intensity of the most extreme wildfires on Earth, Nat Ecol Evol, 8, 1420–1425, https://doi.org/10.1038/s41559-024-02452-2, 2024.

Dennison, P. E., Brewer, S. C., Arnold, J. D., and Moritz, M. A.: Large wildfire trends in the western United States, 1984-2011, Geophys. Res. Lett., 41, 2928–2933, https://doi.org/10.1002/2014GL059576, 2014."

b.  *L39: Regarding GFED and other inventories, the burned area is obtained from satellite-based measurements. Typically, a bottom-up approach is associated with ground-based, local, or detailed data collection at specific sites or sources (such as factories, vehicles, etc.). However, satellite data is more commonly associated with a top-down approach, where broad, large-scale observations (e.g., of emissions, land cover, or atmospheric concentrations) are made, and then models or algorithms are used to estimate emissions, trends, or impacts at finer scales. Please, consider checking the terminology of these approaches.*

We have checked the terminology of top-down and bottom-up approaches as applied to the fire emissions inventories. Multiple sources cited in our manuscript use the same terminology (see Darmenov and da Silva, 2015; Wiedinmyer et al., 2011; and Wiedinmyer et al., 2023). Further, the bottom-up approach is defined by using fuel accounting to determine emissions, while the top-down approach is defined by using energy to determine emissions (see Wiggins et al., 2021).

Response Reference:
Wiggins, E. B., Anderson, B. E., Brown, M. D., Campuzano-Jost, P., Chen, G., Crawford, J., Crosbie, E. C., Dibb, J., DiGangi, J. P., Diskin, G. S., Fenn, M., Gallo, F., Gargulinski, E. M., Guo, H., Hair, J. W., Halliday, H. S., Ichoku, C., Jimenez, J. L., Jordan, C. E., Katich, J. M., Nowak, J. B., Perring, A. E., Robinson, C. E., Sanchez, K. J., Schueneman, M., Schwarz, J. P., Shingler, T. J., Shook, M. A., Soja, A. J., Stockwell, C. E., Thornhill, K. L., Travis, K. R., Warneke, C., Winstead, E. L., Ziemba, L. D., and Moore, R. H.: Reconciling Assumptions in Bottom-Up and Top-Down Approaches for Estimating Aerosol Emission Rates From Wildland Fires Using Observations From FIREX-AQ, JGR Atmospheres, 126, e2021JD035692, https://doi.org/10.1029/2021JD035692, 2021.

c.  *L40: 'as a proxy for amount of material burned'… Please, try to avoid repetition for a more compact writing.*
We have removed the repetition and amended the text as follows:

"There are two distinct approaches for estimating global fire emissions within biomass burning emission inventories: (1) a bottom-up approach that uses burned area  and (2) a top-down approach that uses fire radiative power (FRP) as **proxies** for amount of material burned."

d. *L55-56: Please, include references related to this statement.*
We have amended the text as follows to include references:

"Recent advances in remote sensing and in situ observations have enabled improved assessments of wildfire emissions of reactive nitrogen **(Gkatzelis et al., 2024; Lindaas et al., 2021; Theys et al., 2020)**."

The following references have been added to the References section (Theys already included):

"**Gkatzelis, G. I., Coggon, M. M., Stockwell, C. E., Hornbrook, R. S., Allen, H., Apel, E. C., Bela, M. M., Blake, D. R., Bourgeois, I., Brown, S. S., Campuzano-Jost, P., St. Clair, J. M., Crawford, J. H., Crounse, J. D., Day, D. A., DiGangi, J. P., Diskin, G. S., Fried, A., Gilman, J. B., Guo, H., Hair, J. W., Halliday, H. S., Hanisco, T. F., Hannun, R., Hills, A., Huey, L. G., Jimenez, J. L., Katich, J. M., Lamplugh, A., Lee, Y. R., Liao, J., Lindaas, J., McKeen, S. A., Mikoviny, T., Nault, B. A., Neuman, J. A., Nowak, J. B., Pagonis, D., Peischl, J., Perring, A. E., Piel, F., Rickly, P. S., Robinson, M. A., Rollins, A. W., Ryerson, T. B., Schueneman, M. K., Schwantes, R. H., Schwarz, J. P., Sekimoto, K., Selimovic, V., Shingler, T., Tanner, D. J., Tomsche, L., Vasquez, K. T., Veres, P. R., Washenfelder, R., Weibring, P., Wennberg, P. O., Wisthaler, A., Wolfe, G. M., Womack, C. C., Xu, L., Ball, K., Yokelson, R. J., and Warneke, C.: Parameterizations of US wildfire and prescribed fire emission ratios and emission factors based on FIREX-AQ aircraft measurements, Atmos. Chem. Phys., 24, 929–956, https://doi.org/10.5194/acp-24-929-2024, 2024.**

**Lindaas, J., Pollack, I. B., Garofalo, L. A., Pothier, M. A., Farmer, D. K., Kreidenweis, S. M., Campos, T. L., Flocke, F., Weinheimer, A. J., Montzka, D. D., Tyndall, G. S., Palm, B. B., Peng, Q., Thornton, J. A., Permar, W., Wielgasz, C., Hu, L., Ottmar, R. D., Restaino, J. C., Hudak, A. T., Ku, I., Zhou, Y., Sive, B. C., Sullivan, A., Collett, J. L., and Fischer, E. V.: Emissions of Reactive Nitrogen From Western U.S. Wildfires During Summer 2018, Geophys Res Atmos, 126, https://doi.org/10.1029/2020JD032657, 2021.**"

e. *L78-79: Please, consider rewording this sentence for more clarity.*
We have reworded this sentence as follows for clarity:

" **We evaluate the assumptions inherent to the EMG approach to assess the EMG's utility at deriving emission rates and lifetimes of wildfire smoke plumes. We use a simple one-dimensional (1-D) horizontal model to perform this evaluation.**"

4. *Data and methods*
5. *FIREX-AQ*

a. *The section name refers to the FIREX-AQ campaign. However, the GCAS technical features, the AMF concept, the retrieval methodology and the a priori data are also explained. Please, consider to use different sections to describe the campaign and the methodology-related topics.*
We have rearranged the sections to include a new section, 2.2 GCAS Methodology. The two paragraphs concerning the AMF concept, the retrieval methodology, and the a priori data are now under this section.

b. *L90: Is the second also measuring 'absolute nadir radiance'? Please, clarify.*
Yes, it is. We have amended the text as follows:

"The GCAS instrument is composed of two push-broom spectrometers: the first records the spectrum as absolute nadir radiance in the ultraviolet to visible (UV-Vis), from 300 to 490 nm, and the second records the spectrum **as absolute nadir radiance** in the visible to near-infrared (Vis-NIR) from 480 to 900 nm (Kowalewski and Janz, 2014)."

c. *L93: A reference here for these instruments would be suitable.*
References have been added to the text as follows:

"The GCAS instrument shares similar design specifications with the TROPOspheric Monitoring Instrument (TROPOMI**; Veefkind et al., 2012**) and Tropospheric Emissions: Monitoring of Pollution (TEMPO**; Chance et al., 2013**) instruments."

The references below have been added to the References section:

"**Chance, K., Liu, X., Suleiman, R. M., Flittner, D. E., Al-Saadi, J., and Janz, S. J.: Tropospheric emissions: monitoring of pollution (TEMPO), SPIE Optical Engineering + Applications, San Diego, California, United States, 88660D, https://doi.org/10.1117/12.2024479, 2013.**

**Veefkind, J. P., Aben, I., McMullan, K., Förster, H., De Vries, J., Otter, G., Claas, J., Eskes, H. J., De Haan, J. F., Kleipool, Q., Van Weele, M., Hasekamp, O., Hoogeveen, R., Landgraf, J., Snel, R., Tol, P., Ingmann, P., Voors, R., Kruizinga, B., Vink, R., Visser, H., and Levelt, P. F.: TROPOMI on the ESA Sentinel-5 Precursor: A GMES mission for global observations of the atmospheric composition for climate, air quality and ozone layer applications, Remote Sensing of Environment, 120, 70–83, https://doi.org/10.1016/j.rse.2011.09.027, 2012.**"

d. *L93: Please, explain briefly in which aspects these instruments are similar.*
We have added the following sentence to explain the instrumental similarity:

"**TROPOMI operates in the 310 to 405 nm and 405 to 500 nm bands with a spectral resolution of 0.55 nm and in the 675 to 725 nm and 725 to 775 nm bands with a spectral resolution of 0.5 nm (Veefkind et al., 2012). TEMPO operates in the 290 to 490 nm and 540 to 740 nm bands with a spectral resolution of 0.57 nm (Zoogman et al., 2017)**."

The following references were added to the References section (Veefkind was added in response to the previous comment):

"Zoogman, P., Liu, X., Suleiman, R. M., Pennington, W. F., Flittner, D. E., Al-Saadi, J. A., Hilton, B. B., Nicks, D. K., Newchurch, M. J., Carr, J. L., Janz, S. J., Andraschko, M. R., Arola, A., Baker, B. D., Canova, B. P., Chan Miller, C., Cohen, R. C., Davis, J. E., Dussault, M. E., Edwards, D. P., Fishman, J., Ghulam, A., González Abad, G., Grutter, M., Herman, J. R., Houck, J., Jacob, D. J., Joiner, J., Kerridge, B. J., Kim, J., Krotkov, N. A., Lamsal, L., Li, C., Lindfors, A., Martin, R. V., McElroy, C. T., McLinden, C., Natraj, V., Neil, D. O., Nowlan, C. R., O'Sullivan, E. J., Palmer, P. I., Pierce, R. B., Pippin, M. R., Saiz-Lopez, A., Spurr, R. J. D., Szykman, J. J., Torres, O., Veefkind, J. P., Veihelmann, B., Wang, H., Wang, J., and Chance, K.: Tropospheric emissions: Monitoring of pollution (TEMPO), Journal of Quantitative Spectroscopy and Radiative Transfer, 186, 17–39, https://doi.org/10.1016/j.jqsrt.2016.05.008, 2017."

e.  *L98: The GCAS instrument measures radiance, from which the differential absorption spectra from a gas is deduced. Therefore, the different absorption spectra is not measured, but deduced. Then, I would consider removing the term 'measured'.*
We have removed the term 'measured' as demonstrated in the following amendment:

"By fitting the differential cross-sections of trace gases to the  differential absorption spectra, the trace gas concentrations along the light path can be determined via the Beer-Lambert law."

f.  *L100: A figure showing an example of transmissivity spectra from NO2 and HONO would be illustrative to justify the selection of these fitting windows. Another option could be cite other works that used them.*
We have cited other works that illustrate either the fitting window's use in previous work or works that demonstrate the absorptivity of the compound in the UV-Vis band. Our amended text is as follows:

"The fitting windows for $NO_2$ and HONO are 425 to 460 nm and 345 to 390 nm, respectively. **The $NO_2$ fitting window is within the standard TROPOMI NO2 product window of 405 to 465 nm (van Geffen et al., 2022). HONO absorbs in the UV-Vis spectrum at wavelengths 342 nm, 354 nm, and 368 nm (Stutz et al., 2000).**"

The new references are added to the References section:

"van Geffen, J. H. G. M., Eskes, H. J., Boersma, K. F., and Veefkind, J. P.: TROPOMI ATBD of the total and tropospheric NO2 data products, 2022.

Stutz, J., Kim, E. S., Platt, U., Bruno, P., Perrino, C., and Febo, A.: UV-visible absorption cross sections of nitrous acid, J. Geophys. Res., 105, 14585–14592, https://doi.org/10.1029/2000JD900003, 2000."

g.  *L105: AMF is not only dependent of the wavelength and altitude (first sentence), but also dependent of the solar zenith angle and the viewing zenith angle, among others. You refer to these parameters by mentioning the observation geometry (second sentence).*

*However, consider rewording these sentences to build a more compact sentence where the dependencies of the AMF are not split in 2 sentences. It is not wrong, but I think this correction can provide more consistency in the AMF description.*
We have rewritten these sentences about AMF as follows:

"An AMF is a  quantity representing the effect that the light's path has on retrieval. An AMF depends on **wavelength, altitude,** observation geometry, surface reflectivity, a-priori vertical profiles, aerosols, and other factors that affect the measurement sensitivity."

h. *L108: Please, include a reference for VLIDORT.*
The reference for VLIDORT has now been included in the text:

"In this study, AMFs are calculated using the vector linearized discrete ordinate radiative transfer code (VLIDORT**), version 7.2 (Spurr, 2006)**."

The full reference is below:

"**Spurr, R. J. D.: VLIDORT: A linearized pseudo-spherical vector discrete ordinate radiative transfer code for forward model and retrieval studies in multilayer multiple scattering media, Journal of Quantitative Spectroscopy and Radiative Transfer, 102, 316–342, https://doi.org/10.1016/j.jqsrt.2006.05.005, 2006.**"

i. *L132: Which is the range of the plume altitude? Is there such a big difference that can have an impact on the results or the selected value is well-suited for this purpose? For example, at different altitude values from the same plume, is there a significant difference in the preassure values? Please, justify the assumption taken regarding the altitude.*
The range of the plume's altitude is from 4 km to 5.25 km, equivalent to 630 mb to 544 mb. The selection of the plume's altitude/pressure level impacts the wind speed taken from ERA5. From 550 to 600 mb, there is a shift in wind as depicted below on the first and second images. From 600 mb to 650 mb, there is no shift in the wind direction as depicted below in the second and third images. However, adding the GCAS plume data, we see that the selection of 588 mb, where the DC-8 sampled the plume, is appropriate, as the plume direction follows the wind vectors, in the fourth image. I have clarified in the text that the DC-8 sampling altitude of the plume determined the pressure level chosen:

"Altitude was converted to a pressure level by recording the DC-8 aircraft data at a time when the DC-8 was flying through the smoke plume. **The DC-8 plane's altitude and the plume's altitude were approximated to be at** a pressure level of 588 mb."

ERA5 Winds at 2019-08-16 22 UTC and 550 mb

[Figure]

Wind Speed (m/s)

0.0    2.6    5.2    7.8    10.4    13.0    6.0

Data Min = 4.4, Max = 10.8

ERA5 Winds at 2019-08-16 22 UTC and 600 mb

[Figure]

Wind Speed (m/s)

0.0    2.6    5.2    7.8    10.4    13.0    6.0

Data Min = 4.7, Max = 12.4

6. **ERA5**
   a. *L138: Please, consider to show the pressure ranges presenting first the lower values and then the higher values.*
   We have rewritten this sentence to reorder the pressure ranges as follows:

   "We collected zonal and meridional wind speeds at pressures  every 50 mb from **400** mb to **750** mb **and every 25 mb from 750 mb to 1000 mb**."

7. **GOES FRP**
   a. *L145: When reading this sentence, it seems that the main goal from these satellites are to locate fires. Please, consider rewording.*

The satellites have multiple goals and purposes for their operation. Specifically relating to fires and of interest to this manuscript, these satellites are used to locate fires and retrieve fire characteristics, which was stated in the main text. The sentence structure remains the same, but we now emphasize their retrieval of the fire properties:

"On both satellites, the Advanced Baseline Imager (ABI) uses visible and infrared spectral bands to locate fires and  **retrieve** fire characteristics."

b. *L149: What is GeoMAC? Please, provide some clarification or at least a reference.*
GeoMAC is an internet-based mapping tool that has been collecting and storing data on wildfire perimeters in the contiguous 48 states of the United States and Alaska since 2000. We have expanded on the main text to clarify what GeoMAC is:

"A diurnal profile of the Sheridan Fire's sum-FRP, a sum of all fire pixel FRPs associated with the Sheridan Fire, was generated by selecting GOES data within 4 km of the final fire perimeter from the Geospatial Multi-Agency Coordination (GeoMAC). **GeoMAC is an internet-based mapping tool that stores wildfire perimeters in the contiguous 48 states of the United States and Alaska since 2000.**"

**8. Emission inventories**

a. *L159: In previous text, the angular degree unit was deg. Please, use just one of these units for consistency.*
We thank the author for catching this inconsistency. We have selected the unit to be ° throughout the text.

b. *L160: This sentences is something misleading. Then, this model can provide emission data for NO and HONO or only for NO? Regarding the way it is written, it seems that it says that HONO is a NOx, but it is not.*
We agree that the way the text is currently written, there is some ambiguity in whether HONO is part of $NO_X$. We have amended the text as follows:

"The spatial resolution is 0.25° × 0.25° and provides emission data for $NO_X$ **(**as NO**)**. **Emissions data is not provided for** $NO_2$ or HONO."

c. *L165: Which are the E units? Please, clarify.*
We have added units for all variables, as demonstrated by the additions below:

"…where *i* is a specific compound, *E* is the emissions **(g)**, *A* is the area burned **(m$^2$)**, *B* is the amount of biomass **(kg m$^{-2}$)**, *FB* is the fraction of biomass burned **(unitless)**, and *EF* is the emission factor with units of mass of *i* per mass biomass burned **(g kg$^{-1}$)**."

d. *L170-173: Please, consider a simpler description.*
We have rephrased these couple of sentences to simplify the description:

"**FINNv2.5 can also derive other emission products from total**  non-methane organic gases (NMOG) **using** three commonly used chemical mechanisms: Statewide Air Pollution Research Center Mechanism (SAPRC99**; Carter, 1999**), Model for Ozone and Related chemical Tracers

(MOZART**; Emmons et al., 2020**), and Goddard Earth Observing System with Chemistry (GOES-Chem**; Bey et al., 2001). FINNv2.5 can derive HONO from SAPRC and MOZART, but not from GOES-Chem.** "

e. *L185: Which are the data sources from GFASv1.2? All the data sources from the previous inventories were described.*
We have amended the text to include GFASv1.2 data sources as follows:

"GFASv1.2 is also an FRP-derived emissions inventory **using the MOD14 and MYD14 products from the MODIS instruments on the Terra and Aqua satellites, respectively**."

9. **Analysis methods**
10. **Calculation of HONO and NO2 line densities**
   a. *L196: This can work for gaussian plume shapes as the ones you show in this study. However, if the direction of the plume is extracted from the plume shape, it will not always align with the wind direction. Please, consider this for the discussion.*
   We have added the following discussion point to the main text, as shown below:

   "Subsequently, we rotated the grids by the angle of the  NO$_2$ VCD plume **averaged over time**,  **acting as a proxy of** the mean wind direction (Fig. A1). **In general, the plume direction may not always align with the wind direction. We applied this rotation methodology to the Sheridan Fire because of its near-ideal plume characteristics, a linear plume shape.**"

   b. *L196: Time averaged VCD plume? Please, clarify.*
   We have rewritten this phrase as follows:

   "Subsequently, we rotated the grids by the angle of the  NO$_2$ VCD plume **averaged over time**,…"

   c. *Figure 1: 2 of these panels are already shown in Figure 2. You could consider remove them from here or from Figure 1. If you keep the panels from this figure, you can allude to Figure 1 to refer to the track 14 results.*
   We believe it is kinder to the reader to provide a repetition of panels (b) and (e) for two reasons. First, Figure 1 clearly delineates the steps made to produce the line densities in panels (c) and (f). Without panels (b) an (e), a casual reader may think the integration occurred over panels (a) and (d) from north to south. Second, the side-by-side comparisons of tracks 12, 14, and 18 in Figure 2 make it easier for the reader to compare the VCD magnitudes all in one place.

   d. *Figure 1: Please, consider a more compact caption.*
   The Figure 1 caption has been revised to the following:

   "Figure 1: Image of (a) gridded and (b) rotated track 14 GCAS HONO VCD. (c) HONO line density from track 14.  (d)**-(f)** **Similar to (a)-(c), but for NO$_2$**."

e. *L206: trackS*
There is already the letter "s" in "tracks" to denote the multiple tracks (12, 14, and 18).

f. *Appendix A: In my opinion, it is very difficult to understand this. Please, consider rewording to solve this.*
We have added some background text to Appendix A to help prime the reader for Fig. A1. This text is shown in response to the reviewer's next comment. We have amended the Fig. A1 caption as follows:

"Figure A1:  $NO_2$ VCD plume **averaged over GCAS sampling time**, **with a plume mask applied**. The black line is a **weighted** linear fit of the plume pixels, **where the plume edges have the higher weight**. The angle of the line is displayed."

g. *Appendix A: Please, consider to add some text in this appendix for clarification.*
We have added the following text to clarify the concept depicted in Fig. A1:

"**To calculate the line densities of a sampled smoke plume, the smoke plume needs to be rotated such that the plume axis is parallel to the x-axis. Fig. A1 below summarizes our methodology of determining the plume axis and its rotation angle for all the Sheridan Fire smoke plumes. First, each data point of the regridded $NO_2$ VCDs from all GCAS-sampled tracks (tracks 10-14 and 17-20) are summed and then divided by the number of valid cells (non-NaN cells), producing a temporal average of the plume shape, since the averaging occurs over the different timestamps of the sampled tracks. Second, we calculate the background $NO_2$ VCD by averaging all cells upwind of the fire, Third, we create a plume mask, where only $NO_2$ VCDs that are greater than three times the background $NO_2$ VCD remain. These are the colored pixels in Fig. A1. Finally, we perform a linear regression on the plume $NO_2$ VCD, weighting the regression with the difference between the $NO_2$ VCDs and the maximum $NO_2$ VCD plume pixel. This weighting prioritizes the plume's edges, which define the plume shape. The angle of this linear fit becomes the plume rotation angle we apply to all the sampled plume tracks.**"

**11. Exponentially Modified Gaussian (EMG)**

a. *L217: NOx = NO + NO2. Then, NO2 repeated?*
That is correct. $NO_X$ and $NO_2$ are different quantities. Modelers are interested in quantifying $NO_X$ emissions for their models, but satellites can only retrieve $NO_2$. The Griffin et al. (2021) paper communicated their results for $NO_2$ solely and for $NO_X$, the sum of $NO_2$ and NO.

**12. Results and discussion**
**13. HONO and NO2 plume structure in the Sheridan Fire**

a. *Why there is a better resolution for NO2 maps? Please, clarify.*
There is a better resolution for $NO_2$ maps because the signal-to-noise ratio of HONO is lower than that of $NO_2$. The maximum $NO_2$ is about four times greater than maximum

HONO. However, HONO and $NO_2$ have the same average minimum sensitivities of 0.7 × $10^{15}$ molec $cm^{-2}$.

We amend our text to explain this concept after Fig. 1 is displayed, which is the first time the resolution difference can be noticed:

"**The resolution of the HONO image is lower than the $NO_2$ image because the magnitude of $NO_2$ VCD is approximately four times greater than HONO. This results in a higher signal-to-noise ratio for $NO_2$.**"

b.  L264-265: '... in all three overpasses, HONO and NO2 share local maxima, plume edges, and plume shape'. Please, consider a discussion about this result. What does this mean?
We have added a small discussion about this result as follows:

"Overall, in all three overpasses, HONO and $NO_2$ share local maxima, plume edges, and plume shape. **This means that both chemicals shared similar emission characteristics, were exposed to a consistent dispersal pattern, and traveled together under the same atmospheric conditions.**"

14. **EMG emission rates and lifetimes from the Sheridan Fire**
a.  L274: I would avoid to use the term 'step function' when you are referring to the extracted line densities as they are not extracted according to a known function. Please, consider to use other terms as 'drop sharply' or similar.
We agree that the use of the term "step function" is not appropriate as applied in this paragraph. We have amended the text as follows:

"In Fig. 3e, track 12 appears to  decrease **sharply** in integrated $NO_2$ VCD around 25 km away from the fire center…"

"HONO appears to have the **sharp** decline in integrated HONO VCD that $NO_2$ demonstrates…"

b.  L291-293: This sentence is somewhat confusing in expressing the timeline of events. Please, consider to reword this.
We have rewritten the sentence as follows:

"**Before analyzing the track 18 line densities, it is crucial to note that**  sum FRP **at track 18** is **back to** the background value ."

c.  L294: You could avoid to refer to Fig. 3f if you are already alluded to 'track 14'.
We agree. We have removed referring to Fig. 3f as shown below:

"The $NO_2$ line density maximum for track 18 in Fig. 3g is nearly four times smaller than that of track 14 "

d.  L301: If you show the ratio of emission rates, you should first calculate them individually. Please, consider to include this calculation in the text (or if not, in an appendix).

The emission rates come from the EMG fits like those shown in Fig. 3 and described in Eq. 5, $E_{EMG}$. We have included the calculation in the text as follows:

"…we can take the ratio of the emissions rates of HONO and NO$_2$ (**$E_{EMG,HONO}/E_{EMG,NO2}$**) **from all flight tracks** to explore how reactive nitrogen in wildfire smoke is partitioned and processed."

e. *L301: What do you mean with 'similar sampling biases'? Please, clarify.*
We mean to say that we acknowledge that HONO and NO$_2$ have the same sampling biases as described in lines 204-207. The sweeping bowtie flight pattern will produce the same incomplete and asymmetric samplings in HONO as in NO$_2$ for every flight track, not just the three tracks mentioned earlier. We have amended the text as follows to make this connection for the reader:

"**Acknowledging** that HONO and NO$_2$ have  **the same** sampling biases, **meaning that each GCAS sample of the plume shares the same sampling orientation of the sweeping bowtie**, we can take the ratio…"

f. *L303: '... less HONO is being emitted than NO2….' Is this correct? As a I understand, it makes more sense to me 'a relative decrease of HONO in reference to previous values' instead of 'less HONO is being emitted than NO2'. 'less HONO is being emitted than NO2' is always true as the ratio is lower than 1. Please, clarify.*
We thank the reviewer for catching this error. We agree with your interpretation and have amended the text as follows:

"A decrease in the emission ratio indicates that over time, **there is a relative decrease in HONO in reference to previous values**."

g. *L306: As I understand, constant emission factor ratios should lead to constant emission ratios. To make this clear, I would change here 'emission factor ratio' to 'emission ratio'.*
We agree. We have amended the text as follows:

"If this were true, HONO and NO$_2$ should have  constant emission  ratio**s** over both time and sum FRP."

h. *Figure 4: How you can extract the emission rates of HONO and NO2 from 7 points (Figure 4) regarding that you only have 3 plumes (3 tracks) (L204-207)*
As stated earlier in the manuscript, only three tracks capture the smoke plume core: tracks 12, 14, and 18. However, in this section, we are extracting and reporting emission ratios. Even though the remaining tracks capture the smoke plume at an angle and are asymmetric, both the HONO and NO$_2$ VCDs share the exact same sampling bias, meaning that they are both asymmetric in the same way. While the emission rates themselves will not be accurate, both emission rates were derived from the same sampling bias. Thus, taking their ratio eliminates this sampling bias. We have amended the text as follows to clarify our methods:

" Acknowledging that HONO and NO$_2$ have  **the same** sampling biases**, meaning that each GCAS sample of the plume shares the same sampling orientation of**

**the sweeping bowtie**, we can take the ratio of the emissions rates of HONO and NO$_2$ (**$E_{EMG,HONO}/E_{EMG,NO2}$**) **from all flight tracks** to explore how reactive nitrogen in wildfire smoke is partitioned and processed. **By taking a ratio, the sampling biases in HONO and NO$_2$ VCDs and EMG emission rates cancel out.**"

15. ***An improved EMG methodology: Monte Carlo diurnal 1-D models***

   a. *L353-355: Please, add a reference related to this idea. Why an increase of thermal ouput lead to an underestimation of the emission rate? Please, clarify.*
   We have included a second reference to Wiggins et al. (2020), where Fig. 1 shows the average diurnal cycle of fires from the FIREX-AQ campaign. Many of these fires have peak thermal output, measured as FRP, after 1:30 PM local time. Thus, a satellite like TROPOMI would not observe the maximum thermal output, and would underestimate the emission rate. We have amended the text as follows:

   "Third, wildfires have been observed to increase their thermal output after the 1:30 PM local time that TROPOMI and some other polar-orbiting satellites observe at **(Wiggins et al., 2020)**, and thus daily **EMG** estimates from  **polar-orbiting satellites** may underestimate maximum emission rate."

   b. *L378: From which time? Please, clarify.*
   The summation and division was conducted from 16 August 7 UTC to 17 August 7 UTC, or midnight to 11:59 PM local time. We have added this clarification to the text as follows:

   " **First, the** diurnal FRP shape is obtained by dividing the summed GOES-16 and GOES-17 5-minute FRP product by the GOES total daily FRP **for times 16 August 7 UTC to 17 August 7 UTC, equivalent to 16 August 00:00 to 23:59 local time**."

   c. *L375-383: As I understand, here you are refering to the 4th PECAN 1-D configuration. Please, clarify this in the text.*
   We have clarified this in the text as follows:

   "We run  **this fourth configuration of the** 1-D PECAN**S** model one hundred times…"

   d. *L378: See above where? Please, clarify.*
   We tried to refer to our discussion in the last sentence of the previous paragraph. We see that this is confusing to the reader and have amended the text as follows:

   "**Second,** EMG functions are fit to all appropriate  **GCAS** observations (**greater than forty minutes after fire start and before fire intensity diminishes**)…"

   e. *L379: Where are these ranges coming from exactly?*
   These ranges come from the population of generated EMG fit lifetimes and total emission rates from the second step. The text has been amended as follows:

   "**…greater than forty minutes after fire start and before fire intensity diminishes**) **to create a population of EMG fit emission rates and lifetimes and establish sampling ranges**."

f.  *L3979: As written here, it seems that the lifetime is also included in the division. Please, consider to reword this sentence to better readability.*
We have broken up this long sentence into shorter, step-by-step sentences to expand on the process and improve readability, as demonstrated by our last four responses. The last sentence of this process is as follows:

"**Third,** the ranges of EMG fit **emission rate divided by the fractional FRP (total daily emission rate)**  and **lifetime**  are used to create sampling distributions for a Monte Carlo simulation."

g.  *L375-383: Please, specify at which data this exercise is going to be applied.*
This exercise will be applied to GCAS data for track 12 and 14. We have amended the text as follows:

"The **root mean square error**, **(RMSE),** between the model and the  **GCAS** observations **for tracks 12 and 14**  **are** shown in Fig. 6 for HONO and $NO_2$."

h.  *L381: PECANS*
The text has been modified to, "We run  **this fourth configuration of the** 1-D PECAN**S** model…"

i.  *L375-L383: This paragraph needs further improvement as it deals with complex concepts that need some clarification.*
We believe that the responses to the past 7 comments clarifies these complex concepts.

j.  *L385: Here, the plotted value is R2. You could say that, regarding a R2 value, you could find that it is approximately constant in a combination of lifetime and emission rate values that follow an inverse relationship. However, you cannot say that 'the emission rate has an inverse relationship with lifetime'. Please, reword this sentence.*
We agree with the reviewer. We have amended the text as follows:

"In Fig. 6, **RMSE is approximately constant in a combination of lifetime and emission rate values that follow an inverse relationship** …"

k.  *Figure 6: Why are two y-axis showing the emission rate in two different units? Please, consider just using one and write to the relation between these 2 units in the text (if needed).*
There are two y-axes showing the emission rate in two different units for the benefit of the reader. There are three kinds of audiences that may be interested in this article. First, there is the satellite retrieval group. They inherently work with molecules. Second, there is the emission inventory and modeling group. They inherently work with grams. Third, the audience group that is not within the first or second audience groups. We are not favoring either of the first or second groups by providing an axis for both. We will be keeping both y-axes.

*l.* *Figure 6: There are 2 colorbars that are representing the same value. Please, consider to keep only one.*
We see how the presence of 2 colorbars is busy, distracting, and slightly repetitive. We have removed the coloring of the randomly sampled models and likewise removed the second color bar. As noted by the reviewer and others, we have remade the figure using root mean square error (RMSE) as the fitting metric.

The updated Fig. 6 is now in the manuscript and is shown below:

[Figure]

*m.* *L400: Which are the input values from the TUV Quick Calculator? Please, clarify.*
Upon looking at the previous inputs for the TUV Quick Calculator, we find that there was room for improvement of the inputs used to calculate HONO's lifetime from photolysis. We have selected a new set of values and provided explanations for our selections. The table found below is now included in "Appendix E: TUV Quick Calculator Inputs and Outputs". The following text has been amended in the manuscript:

"The lifetime of HONO from photolysis alone just above the smoke plume, estimated from the TUV Quick Calculator (**Madronich, 2016** ), would be  **11** min and  **13** min during the overpasses of track 12 and 14 respectively, which is shorter than those lifetimes found in Figs. 6a and 6b. **The inputs and outputs of the TUV Quick Calculator can be found in Tables E1 and E2.**"

The input and output values from the TUV Quick Calculator are as follows in the tables below and in the manuscript:

**Table E1. TUV Quick Calculator Inputs**

| Inputs | Values | Justification |
|---|---|---|
| Wavelength Start (nm) | 280 | Default |
| Wavelength End (nm) | 700 | Default |
| Wavelength Increments (-) | 420 | Default |
| Latitude (°) | 34.68 | Sheridan Fire latitude |
| Longitude (°) | -112.89 | Sheridan Fire longitude |
| Date (YYYMMDD) | 20190816 | Analysis Date |
| Time (hh:mm:ss, GMT) | 22:45:15 and 23:20:30 | Track 12 and track 14 ER-2 overpass times |
| Overhead ozone column (du) | 300 | Default |
| Surface Albedo (0-1) | 0.15 | Default albedo of forests |
| Ground Elevation (km asl) | 1.55 | Sheridan Fire elevation |
| Measurement Altitude (km asl) | 6.25 | 4.7 km altitude |
| Clouds Optical Depth (-) | 0 | No clouds |
| Clouds Base (km asl) | N/A | No clouds |
| Clouds Top (km asl) | N/A | No clouds |
| Aerosols Optical Depth (-) | 0.5 | DC-8 DIAL measurement |
| Aerosols Single Scattering Albedo (SSA; -) | 0.7 | Default smoke SSA; 0.37 – 0.95 are valid values (Lewis et al., 2008; Liu et al., 2014) |
| Aerosols Alpha | 1.63 | Saleh et al. (2013) |
| Sunlight Direct beam (-) | 1 | Default |
| Sunlight Diffuse down (-) | 1 | Default |
| Sunlight Diffuse up (-) | 1 | Default |

**Table E2. TUV Quick Calculator Outputs**

| Output Times | HONO photolysis frequency ($s^{-1}$) | HONO photolysis lifetime (min) |
|---|---|---|
| 22:45:15 | 1.467E-03 | 11.36 |
| 23:20:30 | 1.290-03 | 12.92 |

Here are the new references added to the manuscript:

"Lewis, K., Arnott, W. P., Moosmüller, H., and Wold, C. E.: Strong spectral variation of biomass smoke light absorption and single scattering albedo observed with a novel dual-wavelength photoacoustic instrument, J. Geophys. Res., 113, 2007JD009699, https://doi.org/10.1029/2007JD009699, 2008.

Liu, S., Aiken, A. C., Arata, C., Dubey, M. K., Stockwell, C. E., Yokelson, R. J., Stone, E. A., Jayarathne, T., Robinson, A. L., DeMott, P. J., and Kreidenweis, S. M.: Aerosol single scattering albedo dependence on biomass combustion efficiency: Laboratory and field studies, Geophysical Research Letters, 41, 742–748, https://doi.org/10.1002/2013GL058392, 2014.

Saleh, R., Hennigan, C. J., McMeeking, G. R., Chuang, W. K., Robinson, E. S., Coe, H., Donahue, N. M., and Robinson, A. L.: Absorptivity of brown carbon in fresh and photo-chemically aged biomass-burning emissions, Atmos. Chem. Phys., 13, 7683–7693, https://doi.org/10.5194/acp-13-7683-2013, 2013."

n. *L424-426: What does this mean? A presence of a large source of RO2 is realistic? If it is not, what you can say about it? Looking at the plume from Figure A1, it can be seen that there is some influence from the surface. Can the surface have an impact on the retrieval and therefore in the lifetime? Please, clarify.*
Yes, the presence of a large source of $RO_2$ is realistic. A previous chemical modeling study based on actual observations of wildfire smoke indicated that there was a missing sink of $NO_x$ and a missing source of organic nitrates (see text added to the manuscript below).

Regarding the surface influence in Figure A1, we believe the reviewer is either talking about the linear artifacts in the image or the pixels outside of the core plume. For the first point, these linear artifacts are the result of overlapping swath samples with some lingering edge processing effects and averaging the swaths over different time periods. For the second point, we believe that the isolated pixels outside of the plume edge are primarily because we chose a conservative plume mask, which was three times the background value. With a less conservative plume mask, the disconnected pixels rejoin the plume.

However, the surface can impact the retrieval by affecting SCDs and AMFs through surface reflectivity, as mentioned earlier in the manuscript. We cannot determine which way the surface reflectivity impacts the lifetime without performing a larger separate analysis, but generally higher surface albedo causes higher AMFs. However, by dividing the SCDs by the AMF, the VCDs now consider the effect that surface reflectivity has on the slant column retrieval. We would anticipate minimal surface impacts on the retrieval and lifetime determination.

We have added the following discussion to the manuscript:

"However, a simple chemical box model applied to the Sheridan Fire indicates that a large source of organic peroxy radicals ($RO_2$) is needed to drive the $NO_2$ chemical lifetime to the short values we infer from applying the 1-D plume model (with no chemical mechanism) to the GCAS observations. **This is not dissimilar to a hypothesized missing $RO_2$ source seen in the Taylor Creek Fire from 2018 (Peng et al., 2021). In that study, a baseline box model simulation of the Taylor Creek Fire overestimated $NO_x$ and underestimated organic nitrates, therefore missing a $NO_x$ to organic nitrate reaction pathway. By incorporating proxy peroxy radicals with organic nitrate formation pathways, their model better reflected the observed $NO_x$ decay. This has crucial implications for understanding the chemical evolution of wildfire smoke.**"

The following reference has been added to the References section:

"Peng, Q., Palm, B. B., Fredrickson, C. D., Lee, B. H., Hall, S. R., Ullmann, K., Campos, T., Weinheimer, A. J., Apel, E. C., Flocke, F., Permar, W., Hu, L., Garofalo, L. A., Pothier, M. A., Farmer, D. K., Ku, I.-T., Sullivan, A. P., Collett, J. L., Fischer, E., and Thornton, J. A.: Observations and Modeling of NOx Photochemistry and Fate in Fresh Wildfire Plumes, ACS Earth Space Chem., 5, 2652–2667, https://doi.org/10.1021/acsearthspacechem.1c00086, 2021."

16. ***Biomass burning emission inventories underestimate observationally constrained emission estimates***

    a. *L434: Please, consider previous comments regarding the bottom-up and top-down approaches.*

    Please see our response to your comment on manuscript L39. We use bottom-up and top-down appropriately.

    b. *Figure 7: In this section, you deal with the minimum and maximum values of emission rate extracted from track #12 and #14 as representative of the whole day. You called them as 'daily emission rate', which is not true because it does not consider the emission rate at different times during the day. Please, consider to change the terminology and to develop a discussion with this change. Stating that there are biases in reference to daily mean emission from inventories can be misleading.*

    In this section, we deal with the minimum and maximum values of daily mean emissions. In each iteration of the 1-D PECANS model, we extract the total daily emission rate and the lifetime. The total daily emission rate is converted to a daily mean emission rate by multiplying the total daily emission rate by the normalized daily FRP to get a diurnal profile of instantaneous emission rates. This diurnal profile is then averaged to produce the daily mean emission rate plotted in Fig. 7.

    We have added text to clarify this conversion as follows:

    "We compare our Monte Carlo diurnal 1-D model HONO and NO2 emissions with those reported in other biomass burning emissions inventories. **We first convert the total emission rate from the lowest RMSE diurnal 1-D model to a daily mean emission rate. The total emission rate is multiplied by the normalized diurnal FRP profile to get a diurnal profile of instantaneous emission rates. This diurnal profile is then averaged to produce the daily mean emission rate.**"

---

## Author Comment (AC3)

Dear Editor,

*Referee #3's comments are in italic* and our answers to all comments of reviewer #1 are embedded in red. **Bold text** are manuscript additions and  are manuscript deletions.

*This manuscript does a very nice job evaluating NO2 and HONO from the Sheridan fire plume during FIREX-AQ using measurements from GCAS. This manuscript is worthy of publication after some minor revisions.*

***Major comments:***

*It would be helpful if the authors could provide a better motivation for Section 3.3. I am a bit confused. To be clear, I fully understand why there's a need to better understand uncertainties in the EMG method and/or develop a better method, but it's unclear to me exactly how Section 3.3 is doing this. Additional clarification is needed.*

Please see a more detailed response about motivation for Section 3.3 below.

*Errors in the wind speed and direction are a likely contributor to the errors in the EMG fit. From Figure 1 it seems that the rotated plume is not perfectly horizontal, partially due to meandering winds. This will lead to an artificial shortening of the derived lifetimes. In my more detailed comments, I suggested ways to look at this uncertainty.*

We agree that plumes that are not perfectly horizontal will lead to artificially shorter derived lifetimes. Please see a more detailed response further below.

*Also, some more discussion on the aerosol effects on the GCAS retrieval would be helpful. See the Introduction of Cooper et al., 2019 (https://agupubs.onlinelibrary.wiley.com/doi/pdf/10.1029/2019GL083673) for a great discussion on this. This is especially relevant for Section 3.4 when comparing to inventories. I am guessing that the shielding effect (aerosols "shielding" the sunlight from NO2 at lower layers) is more prominent than the enhancement effect (aerosols "enhancing" sensitivity to the GCAS measurement above the wildfire plume layer), and therefore would consistently underestimate NO2 and HONO emissions, but this is merely a hypothesis.*

A discussion of the aerosol effects on the GCAS retrieval is described in more detail below. We thank the reviewer for providing an example of an aerosol discussion.

***More detailed comments are below:***

*Line 87. What was specific altitude of the ER-2? It's relevant because it's important to know how much of the upper troposphere and lower stratosphere is being observed by GCAS. Also can you briefly comment here or elsewhere in the manuscript regarding how much NO2 and HONO is in the lower stratosphere that could have been observed by GCAS?*

The specific altitude of the ER-2 was roughly 20.35 km during the sampling of the Sheridan Fire, so the aircraft was flying in the lower stratosphere. However, as our paper presents the enhanced HONO and NO$_2$ VCDs by removing the upwind background VCDs, and not the absolute HONO and NO$_2$ VCDs, the stratospheric component is removed.

The text has been amended as below:

"On this same day, the ER-2 aircraft flew over the fire in such a way to create a sweeping bowtie pattern **from an approximate altitude of 20.35 km**, where the fire plume was captured on the downwind side of the bowtie and the background air was captured on the upwind side of the bowtie…

We then subtracted from the entire scene the average HONO and $NO_2$ VCDs upwind of the fire**, called the background HONO and $NO_2$ VCDs, resulting in enhanced HONO and $NO_2$ VCDs solely from the wildfire. This process also removes any stratospheric component of HONO and $NO_2$**."

*Line 133. What is the 588 mb pressure level referring to: the plane altitude or the mean plume height or both or neither? Can you clarify? And if it's not the mean plume height, what was the mean plume height? And did this vary by time of day?*

The 588 mb pressure level refers to the plane altitude, but is also an approximation of the mean plume height. The DC8 performed a flight track above the plume, flying along the plume's length and back just after 17 August 2019 0 UTC. The DIAL instrument captured the plume at an altitude of roughly 4700 m. When the DC8 made cross-sectional tracks of the plume, its flight altitude was 588 mb. As the DC8 flew only two lengthwise tracks during the Sheridan Fire, we cannot determine how plume height varied by time of day,

To clarify the main text, we have amended it as follows:

"Altitude was converted to a pressure level by recording the DC-8 aircraft data at a time when the DC-8 was flying through the smoke plume. **The DC-8 plane's altitude and the plume's altitude were approximated to be at**, which resulted in a pressure level of 588 mb."

*Figure 1. As a sensitivity study, it may worthwhile to artificially rotate the plume even further, maybe another 20 degree clockwise to see what effect that has. It does seem that the plume meandered a bit, from southwesterly flow initially to more westerly flow 20 km downwind. I don't know if it's possible, but I would recommend rotating the 0-10 km section of the plume differently than the 10-60 km section of the plume. I think this may be a partial cause as to why the lifetimes from the EMG method are underestimated as discussed in Lines 345-350.*

As mentioned above, rotating the plume even further will shorten the lifetimes due to a compressed line density, and enhance the emission rates since more emissions will be present in a compressed line density. In theory, there should be a wind direction that results in the maximum lifetime and minimum emission rate. This may prove to be a better wind direction estimate, however this method is not perfect either, namely, that nonlinear plumes will not have a clear, best rotation angle based on the wind direction.

We agree that our wind rotation method is not perfect and is made useless under situations where a wildfire smoke plume is not linear, and we acknowledge this limitation in Line 463 of the original submittal.

We do not think that the underestimation of lifetimes as discussed in Lines 345-350 is due to the wind direction. In this idealized 1-D model, there is only one direction for the emissions to go. In theory, the lifetimes should be at their peak characterization. However, it is because of the fast-shifting emission rates that the EMG fits underestimate the lifetime.

*Lines 246 - 249. I'm not following the 3rd and 4th configurations. Is the motivation of this to better quantify the pulsing nature of wildfires? Or something else? One or two more sentences motivating these four sensitivity studies could be very helpful.*

To clarify the motivations between each model configuration, the following text has been added to the manuscript:

"In the first configuration, the emissions rate is kept constant for the entire model**ed** run time **and is the simplest model configuration**. In the second configuration, the emissions rate has a step-change halfway through the model**ed** run time, **adding complexity to the modeled emissions rate**. In the third configuration, the emissions rate is multiplied by the probability density function of a Gaussian with a mean of 5,000 s and standard deviation of 1,000 s**, as daily fire activity has been modeled with a Gaussian distribution previously (Andela et al., 2015)**. Finally, in the fourth configuration, the emissions rate is prescribed such that for every time step, the **total daily emission** rate is multiplied by the fractional FRP**, representing a data assimilation scenario since emission rates have been modeled as functions of FRP**."

The following references were added to the References section:

"**Andela, N., Kaiser, J. W., Van Der Werf, G. R., and Wooster, M. J.: New fire diurnal cycle characterizations to improve fire radiative energy assessments made from MODIS observations, Atmos. Chem. Phys., 15, 8831–8846, https://doi.org/10.5194/acp-15-8831-2015, 2015.**"

*Figure 3a. Can you clarify x-axis to be 16-Aug 16Z, etc. It's not intuitive that the second number is the hour.*

We have edited Fig. 3a as requested by the reviewer. The new figure within the manuscript is shown below:

[Figure]

*Lines 319 - 351. I am not fully following Figure 5 and the associated analysis. I think more detail motivating this section is needed near Line 319. I think you are trying to better understand how the EMG method could better capture the pulsing nature of wildfires, but that's not explicitly said, so maybe I am misinterpreting. Also see comments regarding Lines 246 - 249 which are contributing to my confusion. The times (200 s 5000 s, etc.) are also confusing to me. Are you referring to how much time it takes on your local computer to do the analysis? Can you better define what these mean?*

We have provided text to further motivate our idealized simulations in Section 3.3. That text is provided below:

"We found evidence of this behavior in Fig. 3e. **Additionally, if a fire subsides and dies out, but the plume is still being transported, the EMG fit would still assign a substantial emission rate to the fire. The EMG method in this situation disconnects the fire from the plume, as seen in Fig. 3g. These situations are complex and nonideal, and the EMG method is not suited for them. This then begs the questions of what situations the EMG method is suited for; how long a fire needs to burn before the EMG method provides an accurate result; how quickly the EMG method responds to a change in fire emissions; and how the EMG method stands against a simplified diurnal emission profile.**"

Additionally, we have realized that instead of model run time, we meant modeled run time, the virtual time within the simulations themselves. We have corrected all instances of "model run time" to "modeled run time" where appropriate. The first instance of modeled run time has been amended as below:

"In the first configuration, the emissions rate is kept constant for the entire model**ed** run time **and is the simplest model configuration.**"

*Line 436. A discussion of any potential systematic biases of your MC diurnal 1-D method is warranted here. In particular, the effects of aerosols on the retrievals should be discussed. I am guessing the aerosols could be causing an underestimate in the GCAS retrieval, which would in turn cause derived emissions to be too small. Something to think about and include as discussion.*

We have added the following brief discussion about aerosol impacts at the end of section 3.4:

"**…As discussed earlier, the VCD retrievals from the GCAS instrument are subject to uncertainties, primarily uncertainties in the AMF determination. One major source of AMF uncertainty is the presence of and the characteristics of aerosols in the retrieval columns (Cooper et al., 2019). Aerosols can increase the AMF through scattering; scattering can increase the light path or increase the radiance observed by remote sensing instruments. Aerosols can also decrease the AMF by shielding a compound below a layer from remote sensing instruments or absorbing aerosols can reduce the scattering back towards remote sensing instruments. Wildfire smoke is made up of both scattering and absorbing aerosols and can be present as dilute or concentrated plumes. With both HONO and $NO_2$ enhancements coinciding within the aerosol layer, we hypothesize that aerosols both shield HONO and $NO_2$ deeper within the plume and absorb sunlight. This will likely lead to an underestimation of HONO and $NO_2$ concentrations and therefore an underestimation of emission rates. Thus, the actual emissions may be more on par with the QFED and GFAS inventories…**"

The following references were added to the References section:

"**Cooper, M. J., Martin, R. V., Hammer, M. S., and McLinden, C. A.: An Observation-Based Correction for Aerosol Effects on Nitrogen Dioxide Column Retrievals Using the Absorbing Aerosol Index, Geophysical Research Letters, 46, 8442–8452, https://doi.org/10.1029/2019GL083673, 2019.**"